# Hydrogen and dark oxygen drive microbial productivity in diverse groundwater ecosystems

S. Emil Ruff [1,2,3] ✉, Pauline Humez[1], Isabella Hrabe de Angelis [1,4], Muhe Diao [1], Michael Nightingale[1], Sara Cho [1], Liam Connors [1], Olukayode O. Kuloyo[1], Alan Seltzer [5], Samuel Bowman[5], Scott D. Wankel [5], Cynthia N. McClain [1,6,7], Bernhard Mayer [1] & Marc Strous [1]

Around 50% of humankind relies on groundwater as a source of drinking water. Here we investigate the age, geochemistry, and microbiology of 138 groundwater samples from 95 monitoring wells (<250 m depth) located in 14 aquifers in Canada. The geochemistry and microbiology show consistent trends suggesting large-scale aerobic and anaerobic hydrogen, methane, nitrogen, and sulfur cycling carried out by diverse microbial communities. Older groundwaters, especially in aquifers with organic carbon-rich strata, contain on average more cells (up to $1.4 \times 10^7 \, mL^{-1}$) than younger groundwaters, challenging current estimates of subsurface cell abundances. We observe substantial concentrations of dissolved oxygen ($0.52 \pm 0.12 \, mg \, L^{-1}$ [mean ± SE]; n = 57) in older groundwaters that seem to support aerobic metabolisms in subsurface ecosystems at an unprecedented scale. Metagenomics, oxygen isotope analyses and mixing models indicate that dark oxygen is produced in situ via microbial dismutation. We show that ancient groundwaters sustain productive communities and highlight an overlooked oxygen source in present and past subsurface ecosystems of Earth.

About 2% of Earth's water resources occur as groundwater, of which half is saline and the other half fresh[1]. This subsurface freshwater, represents around 30% of the global freshwater resources, sixty times more than in all lakes, rivers, and the atmosphere combined, and exceeded only by the inaccessible and currently still frozen polar ice caps[1]. Aquifers and rock fractures may also hold up to 30% of the total microbial biomass on Earth[2,3], contribute substantially to carbon fixation[4], and contain high proportions of uncultured archaea, bacteria, and viruses[2,5] with a broad spectrum of lifestyles[6]. Despite the global occurrence of groundwater and the magnitude and diversity of its resident biomass, our understanding of the composition and

activity of microbial communities inhabiting these hidden aquatic ecosystems is still patchy, often developed from samples of a few select wells or a single aquifer. In particular, the geochemical and ecological processes that shape groundwater microbial communities over space and time are not well constrained[7].

Establishing robust links between microbial communities and geochemical characteristics of groundwaters requires large datasets having a comprehensive environmental inventory associated with each microbial community sample. To this end, the Groundwater Observation Well Network (GOWN) maintained by Alberta Environment and Protected Areas (AEPA) in Canada, has compiled

[1]Department of Geoscience, University of Calgary, Calgary, Canada. [2]Josephine Bay Paul Center for Comparative Molecular Biology and Evolution, Marine Biological Laboratory, Woods Hole, MA, USA. [3]Ecosystems Center, Marine Biological Laboratory, Woods Hole, MA, USA. [4]Multiphase Chemistry Department, Max Planck Institute for Chemistry, Mainz, Germany. [5]Department of Marine Chemistry and Geochemistry, Woods Hole Oceanographic Institution, Woods Hole, MA, USA. [6]Alberta Environment and Protected Areas, Calgary, Canada. [7]Alberta Biodiversity Monitoring Institute, Edmonton, Canada. ✉e-mail: eruff@mbl.edu

geochemical data for over 250 groundwaters obtained from monitoring wells in different aquifers and geographic regions, representing a variety of geochemical regimes and groundwater ages. Each GOWN well has been sampled repeatedly for many years, including some for several decades[8]. Since 2006, this comprehensive monitoring program has systematically collected regular water level and chemical water quality information and isotopic compositions for aqueous and gaseous samples[9]. The province of Alberta is situated in the Western Canadian Sedimentary Basin, which hosts major oil, gas, coal, as well as sulfur, salt, limestone, and dolomite deposits[10]. The shallow and deep subsurface has been extensively studied in the context of petroleum and coal exploration and development[11](Fig. 1).

In this work, we analyzed 138 groundwaters from 95 GOWN wells completed in bedrock and surficial aquifers across Alberta (Fig. 1, Table S1, Supplementary Data 1) with the aim of investigating the biogeochemistry and microbial ecology of a broad range of aquifer environments. We employed a multidisciplinary characterization of the groundwater geochemical composition including major and minor ion concentrations, dissolved gases, determination of groundwater ages, and the compositions and metabolic capabilities of its resident microbial communities. The guiding objectives were to determine (i) whether geochemistry is consistent with microbial community compositions and metabolisms, (ii) which electron donors and acceptors support these communities, and (iii) which microbial lineages represent key populations. We show that ancient groundwaters can harbor productive and diverse microbial communities. Metagenome analyses and geochemistry reveal that these communities make a living using hydrogen, methane, sulfur, and molecular oxygen. We show that molecular oxygen was likely produced in situ and plays a role in the biogeochemistry and ecology of these subsurface ecosystems.

## Results and discussion

### Geochemical evolution of groundwater in the Canadian Prairie

The mean residence time of water in an aquifer is a very important factor influencing its geochemical composition (or facies) (Fig. 2a). Mean residence time, or groundwater age, is determined using radioisotopes such as tritium and $^{14}C$ and refers to the travel time between the point of infiltration and the point of sampling[12]. In this study, groundwater age inversely correlated with the ratio between calcium and sodium concentrations in groundwater (Fig. 2b). The youngest, tritium-containing groundwaters (indicating recharge after 1952) were characterized by low total dissolved solids ($<400\,mg\,L^{-1}$) and high Ca/Na ratios (median: 3.5). Dissolution of carbonates during infiltration led to high calcium, magnesium, and bicarbonate concentrations in these waters. These young groundwaters (yellow symbols, Fig. 2) generally had low methane and sulfate concentrations (Fig. 3a, b) and were predominantly collected from wells completed in surficial Neogene-Quaternary deposits (Fig. 1b).

In contrast, the oldest groundwaters (purple symbols, Fig. 2) contained no tritium and little $^{14}C$, indicative of groundwaters more than several hundreds or thousands of years old. They had elevated total dissolved solids ($>900\,mg\,L^{-1}$) and a low Ca/Na ratio (median: 0.01). Old groundwaters were characterized by reducing conditions and contained high dissolved methane concentrations ($12.8 \pm 2.4\,mg\,L^{-1}$ [mean ± SE]; median: $0.72\,mg\,L^{-1}$, range: $0.001$–$74.2\,mg\,L^{-1}$; Fig. 3a, Supplementary Data 1). These waters had elevated sodium, bicarbonate, and chloride concentrations resulting

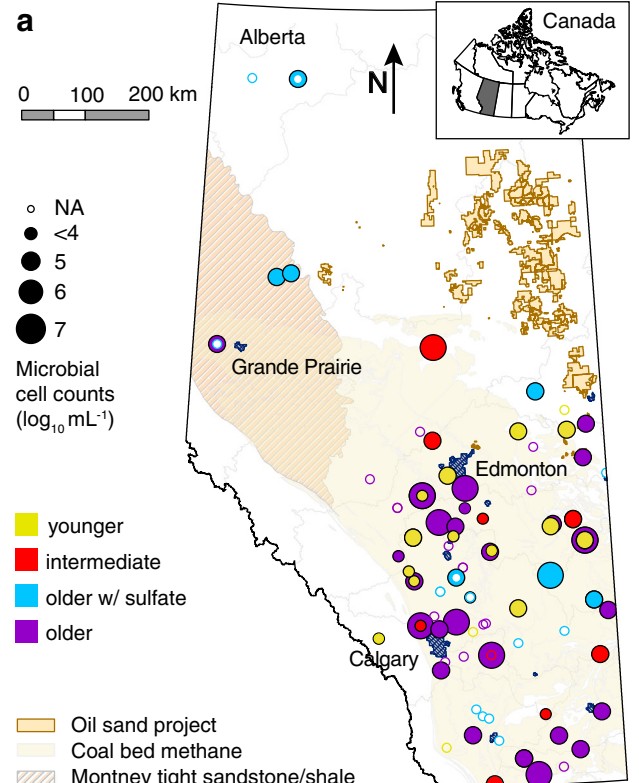

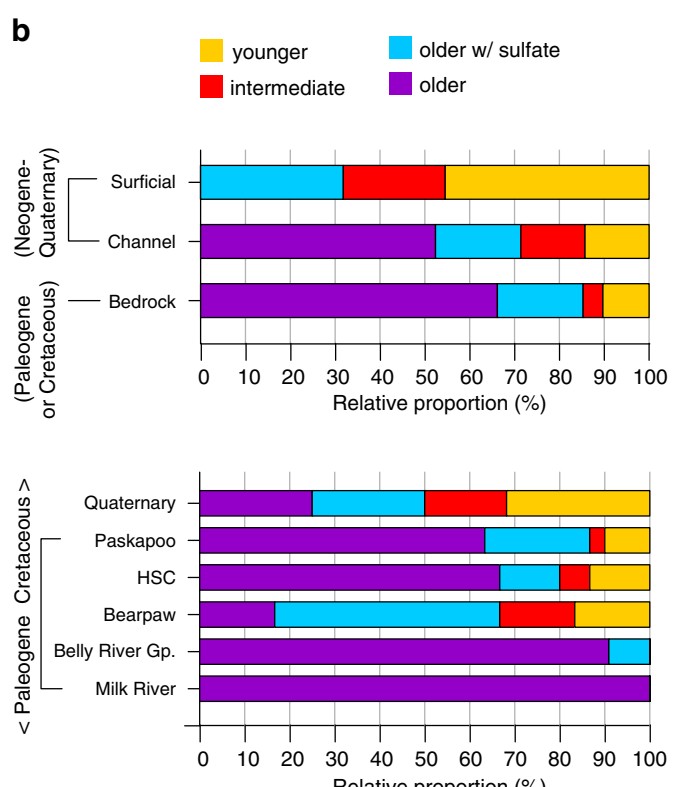

**Fig. 1 | Sampling locations, geological formations, and groundwater ages.**
**a** Location of studied groundwater wells within the energy resources context of the province of Alberta. Colors indicate the groundwater age at each well (yellow: younger waters; red: intermediate age; blue: older waters that are sulfate-rich; purple: older waters with little sulfate). Circle size represents average microbial cell numbers in the groundwater samples, ranging from $10^4$ (smallest full circle) to $10^7$ cells per mL (largest circle). The map was created using Arc-GIS v10.8 **b** Relative proportion of water types in the surficial, channel, and bedrock sediments, as well as in major geological formations of Alberta, showing that groundwater geochemistry evolved with the increasing age of the formations. *NA* not assessed, *HSC* Horseshoe canyon, *Gp.* Group.

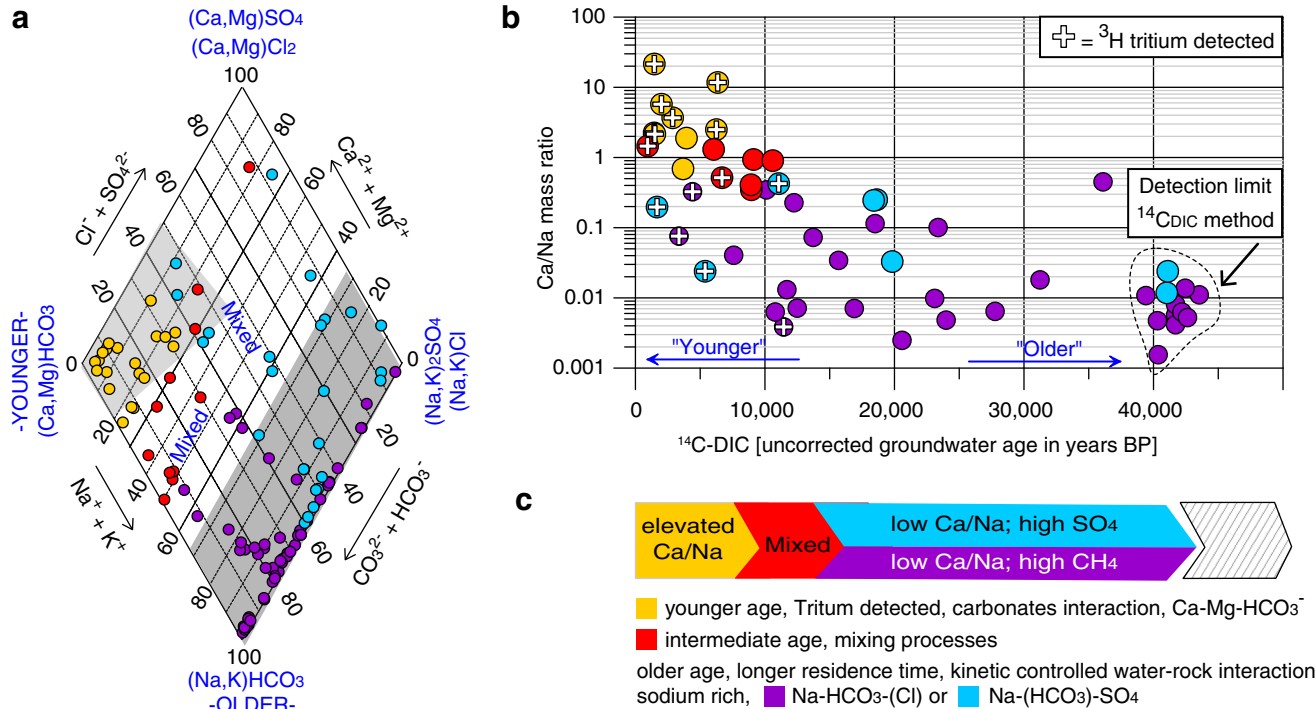

**Fig. 2 | Groundwater geochemistry and age dating. a** Piper diagram of hydro-chemical facies of each groundwater sample (circles) visualizing calcium/sodium mass ratio used as a proxy for geochemical evolution, i.e., residence time/age. Na: sodium, K: potassium, Ca: calcium, Mg: magnesium, Cl⁻: chloride, SO₄²⁻: sulfate, CO₃²⁻: carbonate, HCO₃⁻: bicarbonate **b** Ca/Na ratio decreases with increasing residence time ($^{14}C_{DIC}$ uncorrected age, BP: before present). $^{14}C$: $^{14}$carbon, DIC: dissolved inorganic carbon. Samples that accumulate in the plot around 40k yrs BP (detection limit of method) may be much older. $^{3}H$-positive samples (circles with crosses) comprise meteoric water past 1952 (H-bomb) and corroborate the age trend. **c** Schematic timeline and summary of water aging.

from water-rock interactions, including ion exchange, and weathering of minerals. The older groundwater samples were obtained from wells completed in buried river valleys (channels) and Paleogene and Cretaceous sedimentary bedrock formations that are often characterized by the presence of coal and/or shale[13].

Sulfate, predominantly derived from pyrite oxidation and to a lesser extent from anhydrite or gypsum dissolution[14], was ubiquitous in a third group of groundwater samples (blue symbols, Fig. 2). These waters were characterized by long residence times, contained even higher dissolved solids (>1700 mg L⁻¹) and had intermediate Ca/Na ratios (median of 0.12). Sulfate was often the most abundant anion and electron acceptor in this group of groundwaters, resulting in sulfate-rich hydrochemical facies with low methane concentrations. These groundwater samples were collected from wells completed in surficial deposits, but also from bedrock aquifers completed in clastic, often marine sedimentary rocks of the Bearpaw formation (Fig. 1b).

The mixing of groundwater of different ages and geochemical water types (Fig. 2a) often results in characteristic intermediate hydrochemical facies (red symbols, Fig. 2). These waters were mostly obtained from aquifers in surficial deposits and occasionally from bedrock aquifers (Fig. 1b). A detailed characterization of the geologic formations, groundwater geochemistry, and ages are provided in the Supplementary Results and Supplementary Data 1.

**Old groundwaters contain biomass-rich microbial communities**

Microbial cell density commonly decreases with increasing depth in marine[15] and terrestrial[3] subsurface ecosystems. This decrease in cell numbers and biomass is often attributed to energy limitation in subsurface ecosystems[16]. Within aquifers cell numbers may not necessarily decrease with increasing depth[17], but due to other factors including increasing water age[18,19]. To our surprise, we found significantly more cells in aquifers with old groundwater than in those with younger water

(Fig. 4, Supplementary Data 2). This suggests that the older aquifers and geochemically evolved groundwaters are in fact productive ecosystems that provide energy for the growth of microorganisms. Average cell numbers in the geochemically evolved groundwater samples reached 10⁷ cells mL⁻¹ (Fig. 4). Overall, microbial cell numbers across the entire studied region, did not decrease with depth (Supplementary Fig. 2a, b). Cell counts were on average slightly higher in aquifers in geologic strata that contained shales or coal (Supplementary Fig. 2c), suggesting that elevated contents of organic carbon may provide additional energy sources to microbes. It should be noted that the cell numbers presented here are conservative estimates likely capturing only the free-living cells from water samples which cannot account for the substantial number of cells living in biofilms[20].

The morphology and size of cells varied greatly, and included cocci-, rod-, vibrio-, and spiral-shaped cells, as well as filaments and small aggregates (Supplementary Fig. 3). Cell morphologies, sharp cell boundaries and the bright signal from nucleic acid staining suggested a largely active community[21]. Average cell size was similar across groundwater samples and hence high cell numbers are a proxy for high biomass and productivity (Supplementary Fig. 3). To the best of our knowledge this is the first time that consistently high cell abundances were documented in old aquifers across a large geographic area (>210,000 km²).

The increase in cell numbers with groundwater age was accompanied by a substantial decrease in archaeal and bacterial diversity (Fig. 5a, b; Supplementary Fig. 4) and a shift in microbial community structure from young to old groundwaters (Fig. 5c, d; Supplementary Fig. 5). Of the environmental parameters that we tested, the ones that most strongly explained the variance in microbial community structure were methane concentrations and geochemical proxies for groundwater age, including the concentrations of sodium, calcium, and magnesium and well depth (Fig. 5e, f). Decreases in diversity and

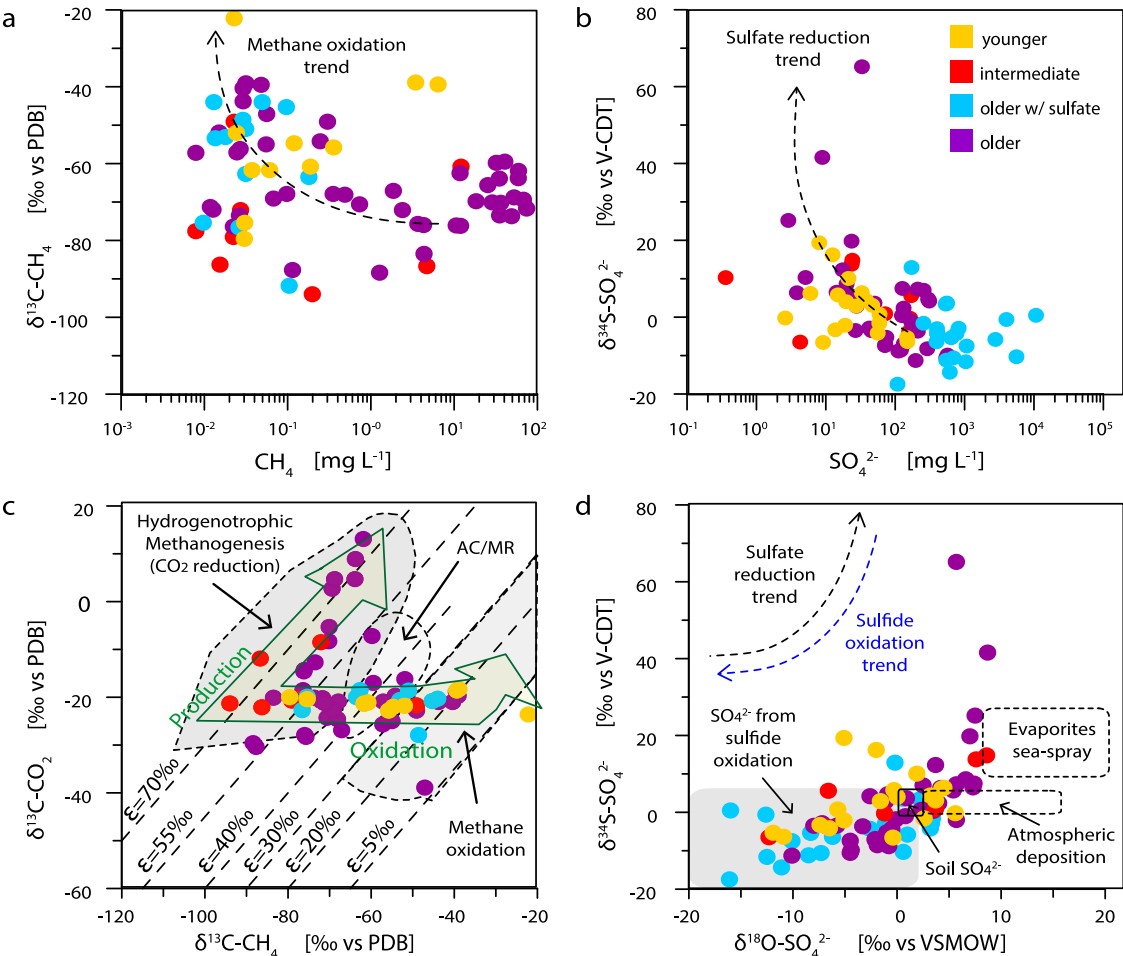

**Fig. 3 | Groundwater isotope geochemistry.** Microbes mediating methane ($CH_4$) and sulfate ($SO_4^{2-}$) cycling impact the carbon and sulfur pools in the groundwater systems. Each circle represents a sample. Circle color depicts water age. Trends concerning reduction and oxidation of the compounds are indicated by arrows or areas. **a** Carbon isotopic signature of methane versus methane concentration. *PDB* Pee Dee Belemnite. **b** Sulfur isotopic signature of sulfate versus sulfate concentration. *V-CDT* Vienna-Canyon Diablo Troilite **c** Carbon isotopic signature of carbon dioxide versus carbon isotopic signature of methane. $\varepsilon$ fractionation $CO_2$-$CH_4$. *AC/MR* Aceticlastic- /methylotrophic methanogenesis. **d** Sulfur isotopic signature of sulfate versus oxygen isotopic signature of sulfate.

shifts in community structure are common features of microbial bloom situations in productive ecosystems, which often feature elevated abundances of a few key species[22,23].

### Hydrogen as basal energy source in productive aquifers

Our results indicated that a major energy source driving high productivity in mature groundwater is hydrogen. Considerable amounts of dissolved hydrogen were detected at selected sites (0.002, 0.007, and 0.1% in samples GW121, GW223, and GW951), and in many aquifers hydrogenotrophic methanogens affiliating with *Methanobacterium*, *Methanoregula,* and *Methanospirillum* showed high relative abundances of 16S rRNA gene amplicon sequence variants (ASVs; Fig. 6a, Supplementary Figs. 6, 7, Supplementary Data 3). Hydrogen serves as sole electron donor for these lineages and the isotope composition of $CH_4$ and $CO_2$ also supports widespread hydrogenotrophic methanogenesis (Fig. 3c). Methane with concentrations ranging from 0.006 to more than 74 mg $L^{-1}$ was present in all analyzed groundwater samples (Fig. 3a). Most samples, however, had concentrations <1 mg $L^{-1}$ (73%; *n* = 106, Supplementary Fig. 6). Based on combined carbon and hydrogen isotope composition, the low methane concentrations likely reflect high relative consumption by microbial methane oxidizers (Fig. 3a).

Among the detected methane oxidizers were obligate anaerobic methane-oxidizing archaea (ANME) of the genus *Methanoperedens*

(ANME-2d, Fig. 6a). These methanotrophs were often found in aquifers that contained either dissolved iron or manganese or both. ANME-2d were found in more than half of the archaeal ASV datasets (34 of 64), predominantly in methane-rich and sulfate-poor older groundwaters, in line with their previously observed occurrence in deep granitic water environments[24]. Their ability to oxidize methane using iron and manganese oxides[25] could explain the high relative ASV abundance of over 40% in the sulfate-depleted mature groundwaters of at least four samples (GW111, GW218, GW311, and GW456). *Methanoperedens* was the fifth most abundant archaeal genus in the studied aquifers and has the metabolic potential to connect the carbon, nitrogen, and metal cycles via the anaerobic oxidation of methane[26,27]. Anaerobic oxidation of methane coupled to bacterial sulfate reduction might occur in certain aquifers as well, because ANME-2ab made up almost half of the archaeal community at GW160, and ANME-2c were detected at GW220 (Supplementary Data 3). Bacterial *Methanomirabiliales*, an order known to comprise microbes that are capable of oxidizing methane in anoxic environments using nitrite[28], were abundant in many aquifer datasets, especially in young and intermediate waters (Supplementary Fig. 8).

Hydrogen also could have sustained the growth of organisms affiliating with the genera *Desulforudis* and *Desulfomicrobium* and many other sulfate and sulfur reducers, and sulfur disproportionators, which were diverse and widespread in the studied groundwater

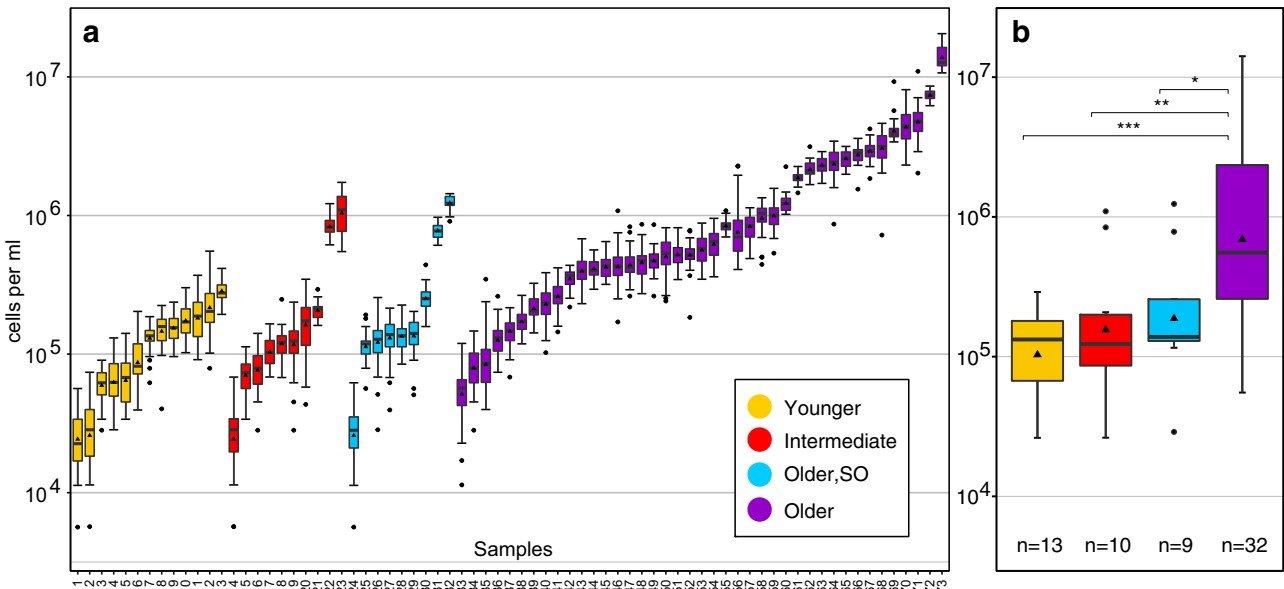

**Fig. 4 | Cell abundances in groundwaters.** Cell abundance was determined using fluorescence microscopy and is given on a logarithmic scale. Boxplots show 1st and 3rd quartile, maximum and minimum values (whiskers), median (line), mean (triangle), and outlier (dots). **a** Boxplots represent groundwater samples from individual wells and summarize cell counts from *n* independent fields of view. *n* = 40: Samples 3–10, 12, 15–22, 24–29, 34–47, 49–51, 54, 56, 63, 67–68, 70–71. *n* = 39: Samples 1, 14, 33. *n* = 38: Sample 2. *n* = 20: Samples 13, 31–32, 55, 57, 61, 72. *n* = 12: Sample 73. For more details on individual cell counts see Supplementary Data 2. *X* axis: Numbers correspond to well IDs (due to space constraints well IDs are given in Supplementary Data 1). Wells which were sampled at two different time points are marked with a star (*), technical replicates are marked with a plus (+). **b** Boxplots represent cell numbers averaged across groundwater age summarizing the average cell numbers in **a** (excluding technical replicates). *n* number of samples. Significance was tested using a Wilcoxon rank sum test. Significance levels are: *$p < 0.05$; **$p < 0.01$; ***$p < 0.001$; (uncorrected).

ecosystems (Fig. 6b, Supplementary Fig. 9, Supplementary Data 4). *Candidatus* Desulforudis audaxviator were the most abundant sulfate reducers across all ASV datasets, yet predominantly occurred in old groundwaters. They often made up >50% of the clades known to comprise sulfur-cycling microbes. *Ca.* Desulforudis are hydrogen-oxidizing, sulfate-reducing *Clostridia* reported to thrive in deep terrestrial aquifer ecosystems[29]. In samples GW3026 and GW217 we found high sequence abundances of microbes affiliating with obligately syntrophic *Smithella* sp. and *Syntrophus* sp. that live together with organisms scavenging hydrogen[30,31].

Shotgun metagenomics supported the inferences derived from amplicon sequencing and revealed genes encoding Ni-Fe hydrogenases, hydrogenotrophic methanogenesis and sulfate reduction in the groundwater communities (Fig. 7). A high-quality metagenome-assembled genome (MAG) of a population within the *Methanobacteriaceae* (UBA349, MAG-32) contained a complete methyl-coenzyme M reductase (*mcrABCG*) and all other genes needed for hydrogenotrophic methanogenesis (*cdhA-E, ftr, mch, fae*), yet lacked genes diagnostic for methylotrophic, or aceticlastic methanogenesis (Supplementary Data 5). A complete sulfate reduction pathway was found in MAG-18 an organism in the genus-level clade UBA2262 within the *Desulfovibrionaceae* (Fig. 7). The hydrogen could be derived from microbial fermentation, e.g., via Fe-Fe hydrogenases that were present in three MAGs affiliating with *Bacteroidia* (*Lutibacter*, UBA6024, JAAYJT01; Supplementary Data 5). These MAGs also encoded the TonB/SusC transport system, which many *Bacteroidia* use to consume oligosaccharides. Hydrogen can also derive from abiotic sources including shales, coal beds, and other organic-rich strata[32], water-rock reactions[33], pumping equipment or steel well casings[34], or even water radiolysis[35,36].

**Aquifers harbor microbes affiliating with aerobic and facultatively anaerobic lineages**

We found high relative ASV abundances of microbes affiliating with obligate and facultatively aerobic hydrogen-, methane-, and sulfur-oxidizing lineages in most of the old groundwaters obtained from confined aquifers (Figs. 6, 8, Supplementary Figs. 7–10, Supplementary Data 4). The presence of potentially aerobic organisms was not the result of sample handling, contamination, or of microbial growth after sampling, and alpha and beta diversity, composition, and cell counts did not change with sample storage time (Supplementary Fig. 11, Methods). Thus we are confident that these microbes were present in the aquifers at the time of sampling. Analyses of metagenomes from five wells (GW114, GW144, GW218, GW265, and GW972, all bearing old groundwaters) corroborated the community composition detected via metabarcoding. Mapped metagenomic 16S rRNA gene short reads (Supplementary Fig. 12), as well as reconstructed full-length 16S rRNA gene sequences (Supplementary Data 6) showed the same key genera and often the same species as the ASV datasets. The reliability of the ASV-based community composition was further supported by very high sequence similarities between ASVs and reconstructed full-length 16S rRNA genes (Supplementary Fig. 13, Supplementary Data 7). Both sequencing approaches show the same key players, including *Methylobacter tundripaludum* (MAG-01 – 04), *Methylotenera* sp. (MAG-09 – 13), *Hydrogenophaga* sp. (MAG-14 – 17), *Sulfuricurvum* sp. (MAG-19) and *Thiobacillus* sp. (MAG-20; Fig. 7).

We detected *Hydrogenophaga* at high relative abundances in most aquifers (Figs. 6b, 7, 8a). *Hydrogenophaga* MAGs contained genes encoding hydrogenases, oxidases, and RuBisCO, but also a sulfur oxidation pathway (*sqr, fccAB, soeABC, soxABX*), a complete denitrification pathway (in 2 of 4 MAGs) and an aerobic carbon monoxide dehydrogenase (*coxSML*), suggesting a facultatively anaerobic lithoautotrophic lifestyle. Microorganisms affiliating with aerobic or denitrifying methano- and methylotrophic bacteria[37] were also abundant in all aquifers, especially in older groundwaters (Fig. 6b, Supplementary Fig. 8). Methylotrophic *Methylotenera* sp. were the most abundant populations in the ASV datasets, followed by methaneoxidizing members of the genus *Methylobacter*. Both genera were also

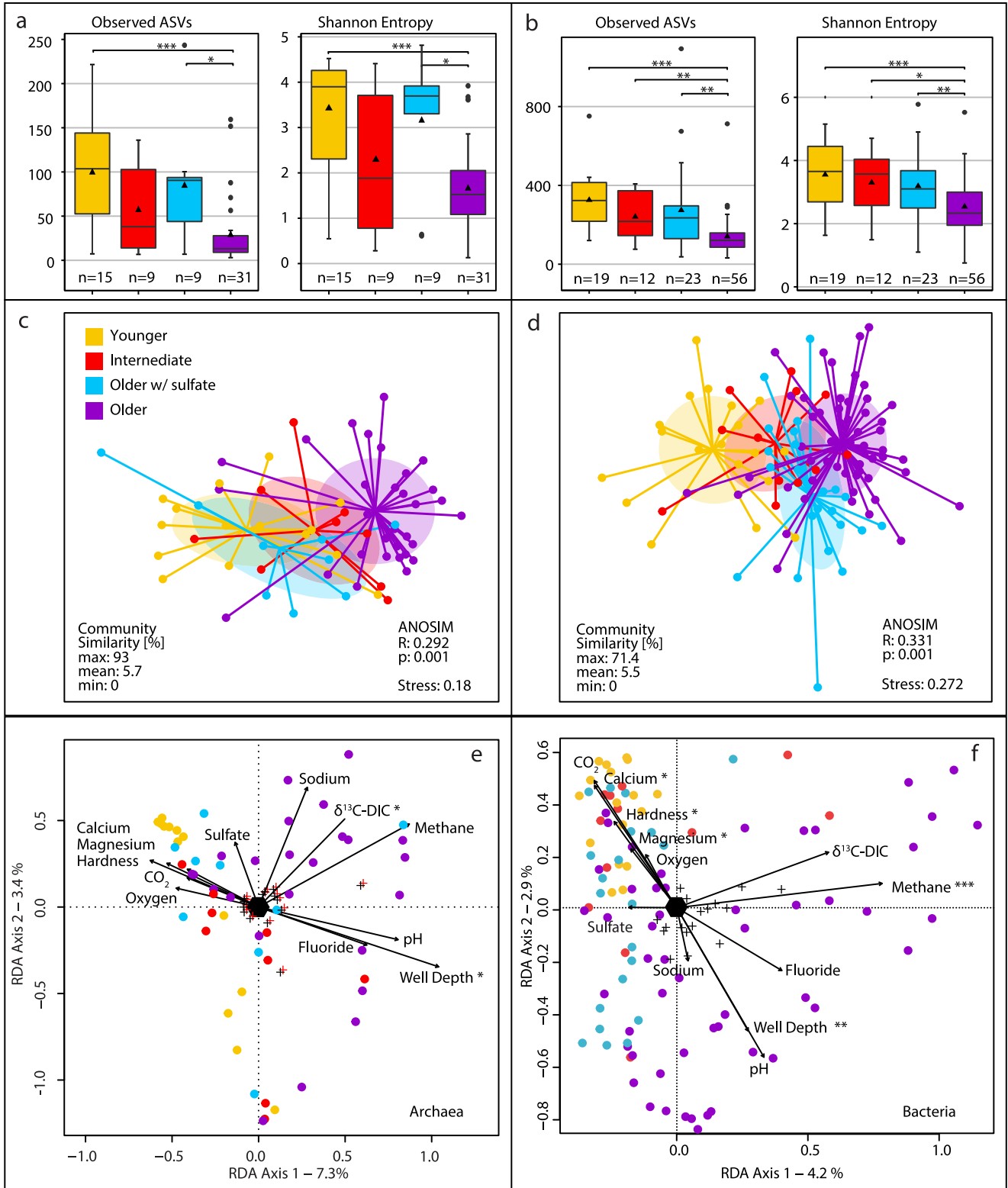

**Fig. 5 | Microbial community diversity. a** Archaeal and **b** bacterial alpha diversity indices of the investigated groundwaters based on 16S rRNA gene amplicon sequence variants (ASVs). Significance was tested using a Wilcoxon rank sum test. Significance levels: \*$p < 0.05$; \*\*$p < 0.01$; \*\*\*$p < 0.001$; (uncorrected). Archaeal **c** and bacterial **d** community structure shown as nonmetric multidimensional ordinations based on community distances. Samples (circles) are connected to the average weighted mean of within group distances (centroid; ellipses show one standard deviation). *n*: see **a**, **b**, respectively. Redundancy analysis (RDA) of **e** archaeal ($n = 64$) and **f** bacterial community structure ($n = 110$) in groundwater samples (circles) using select parameters (arrows). Significance levels: \*$p < 0.05$; \*\*$p < 0.01$; \*\*\*$p < 0.001$; (uncorrected). The full model was highly significant for both domains, and together the 11 parameters explained 11% of archaeal and 18% of bacterial variation. The color legend in **c** applies to all panels. *CO₂* carbon dioxide, *DIC* dissolved inorganic carbon.

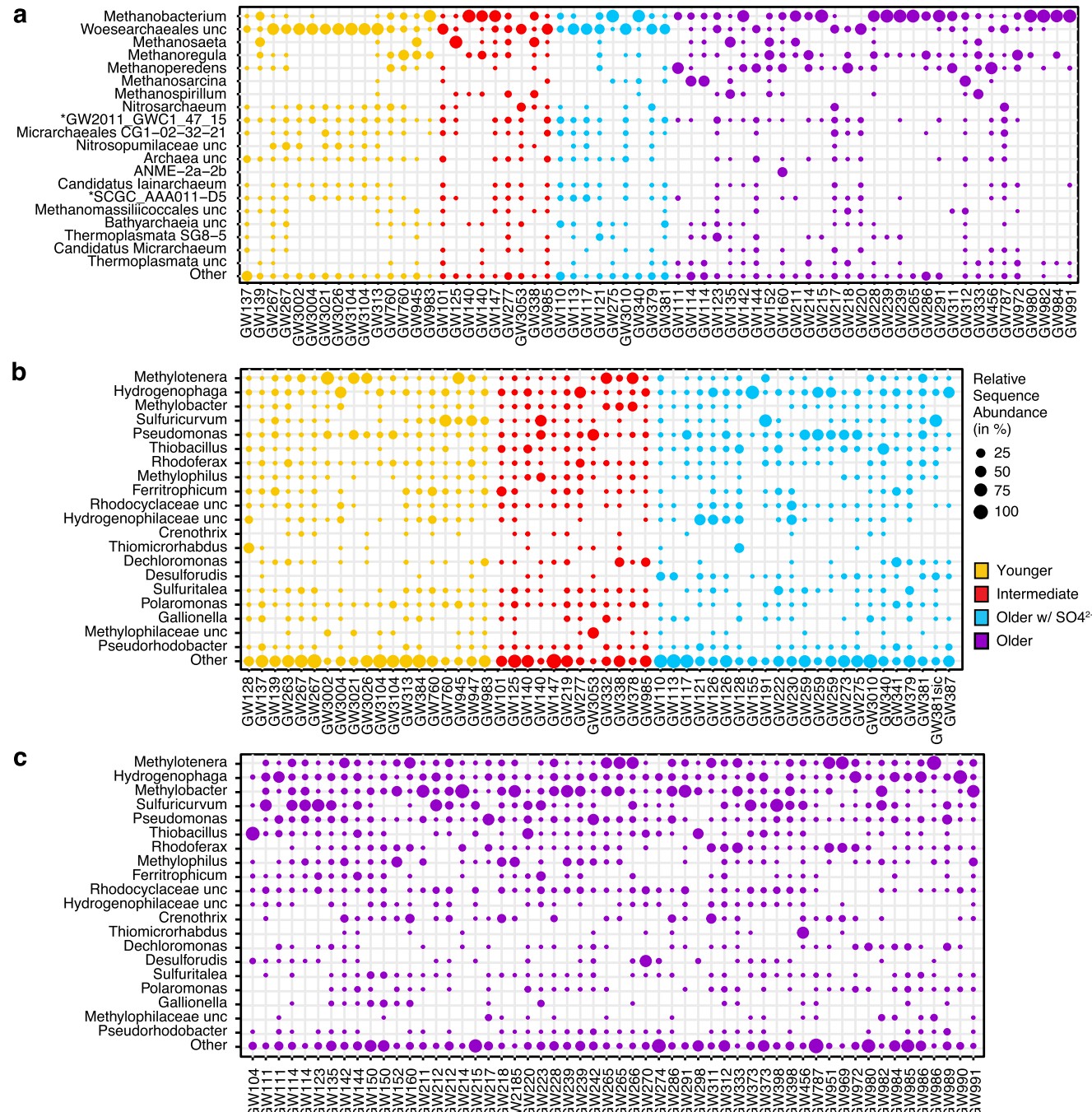

**Fig. 6 | Microbial community composition.** Relative sequence abundance of the top 20 most abundant **a** archaeal and **b, c** bacterial lineages at genus level based on 16S rRNA gene amplicon sequence variants. The remaining lineages with lower abundances are summarized as Other. Genera that were unclassified in the SILVA reference database are abbreviated as unc and the next higher phylogenetic level is shown. The two archaeal clades denoted by * belong to the order *Woesearchaeales*. One methanogen, as well as almost half (9 of 20) of the top bacterial genera are also represented by metagenome-assembled genomes (Fig. 7). The legend in **b** applies to all panels.

abundant in the metagenomes of the five wells and were represented by nine MAGs. *Methylobacter* MAGs had complete pathways for the oxidation of methane to carbon dioxide including the critical activation of methane with molecular oxygen via the particulate methane monooxygenase (*pmoABC*). It was recently suggested that *Methylobacter* can couple methane oxidation to denitrification in hypoxic conditions[38]. Indeed, we found these organisms to be capable of partial denitrification, from nitrite to nitrous oxide. *Methylotenera* MAGs contained genes to oxidize methanol to carbon dioxide and reduce nitrate to nitrite (Fig. 7). Denitrification by *Methylotenera* was reported

previously[39] and hence *Methylotenera* could provide nitrite to *Methylobacter*, while *Methylobacter* provides methanol to *Methylotenera*. Complementary interactions between *Methylobacter* and *Methylotenera* were observed in a methane-rich shallow aquifer[40], a lake[41] as well as in methanotrophic enrichment cultures[42]. The metabolic capabilities of the consortium however suggest that molecular oxygen needs to be available to catalyze the initial activation of methane, even if subsequent steps can be coupled to denitrification. *Methylicorpusculum* sp. (MAG-05), in contrast seemed to be entirely lacking the capability to use other electron acceptors than oxygen (Fig. 7).

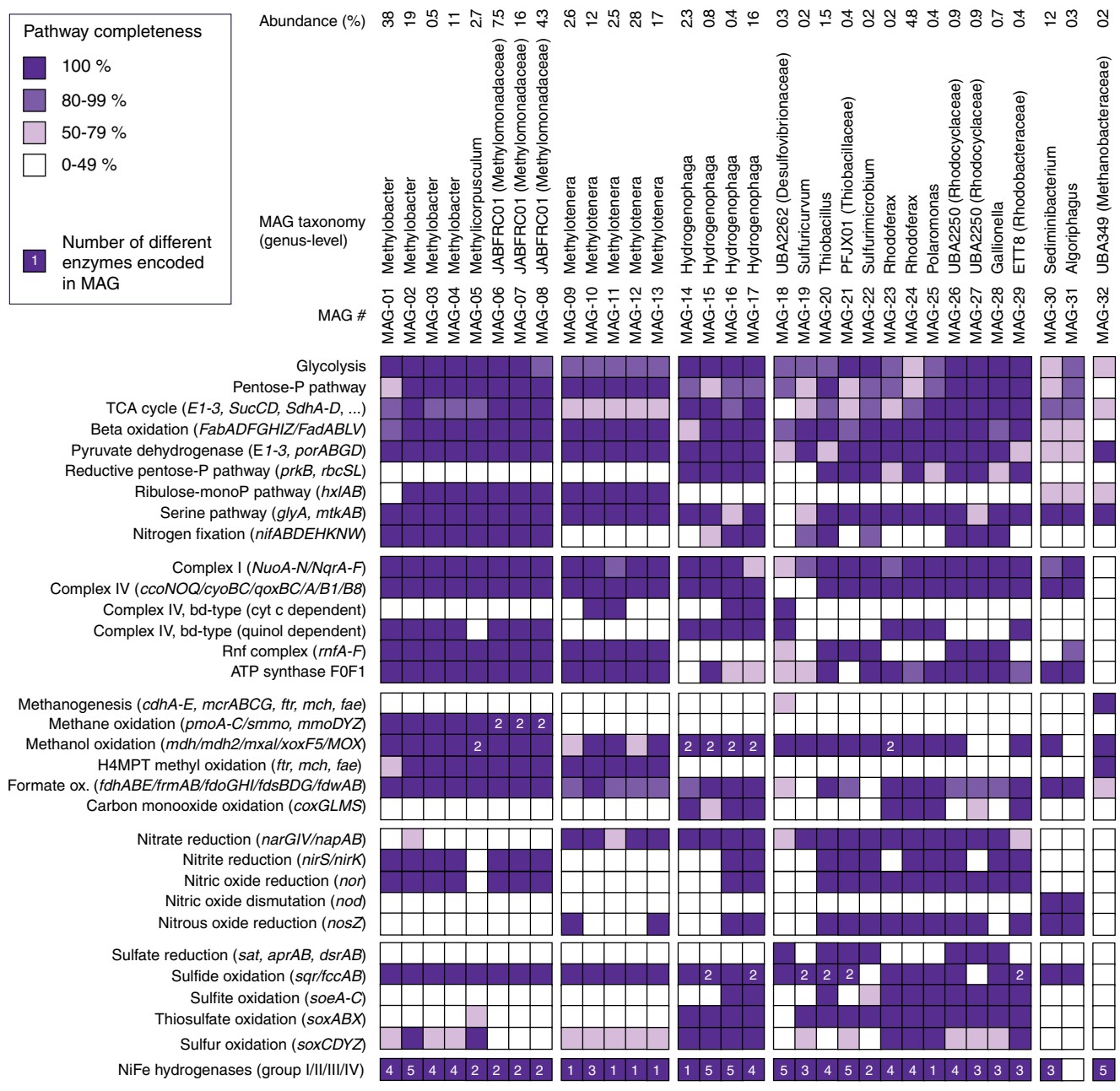

**Fig. 7 | Metagenome-assembled genomes and key metabolic pathways.** Selected metagenome-assembled genomes (MAGs) of abundant microbial populations in groundwaters show the genetic capabilities of methane, sulfur, and nitrogen cycling. Color shade depicts the completeness of the pathway, the included genes are shown in brackets. Numbers refer to redundancy in a pathway or enzyme. As an example, MAG-06 has two methane monooxygenases, one encoded by *pmoABC*

the other by *smmo* and *mmoDYZ*. Commas between genes: and; dashes between genes: and/or. MAG taxonomy is based on GTDB-Tk, and abundance was estimated by read mapping considering binned and unbinned reads. The full set of 61 metagenome-assembled genomes and 450 annotated genes is provided in Supplementary Data 5.

Aerobic methane oxidizers affiliating with the genus *Crenothrix*[43] were the fourth most abundant methanotroph and occurred in almost half of the ASV datasets (45 of 109, Supplementary Fig. 8, Supplementary Data 6). *Crenothrix* sp. are filamentous microbes that can occur in groundwater[44] and may represent the filamentous cells observed with the microscope in many samples (Supplementary Fig. 3). We also retrieved three high-quality MAGs of the uncultured genus JABFRC01 within the *Methylomonadaceae* (MAG-06 – 08). These MAGs encoded a soluble and a particulate methane monooxygenase and complete methane oxidation pathway revealing that the organism is an aerobic methane oxidizer requiring oxygen for the initial oxidation of methane to methanol. Like the closely related *Methylobacter* the organisms

affiliating with JABFRC01 may use nitrite as an alternative electron acceptor for the further oxidation of methanol to carbon dioxide in hypoxic conditions (Fig. 7, Supplementary Data 5). Of the twenty most abundant bacterial genera known to oxidize one-carbon compounds (Supplementary Fig. 8), twelve were associated with oxidation of methanol, methylamines, and other methyl compounds[45], while eight were associated with methane oxidation[46]. The widespread presence of microorganisms that are capable of aerobic methanotrophy corroborates previous findings of aerobic methanotrophic activity in coalbed formations[47,48], with our study expanding this finding to aquifers in more diverse geological settings and a much larger geographical scale.

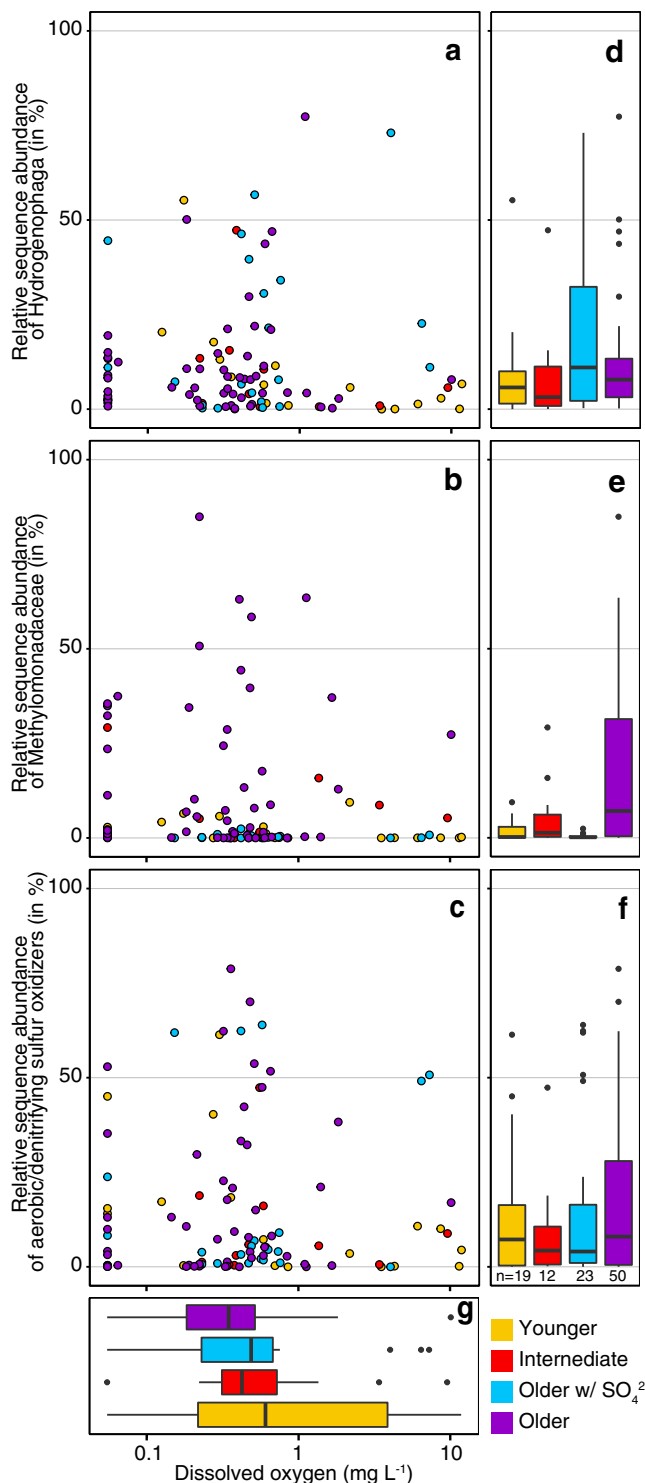

**Fig. 8 | Abundance of aerobic/denitrifying hydrogen, methane, and sulfur oxidizers versus oxygen concentration. a** *Hydrogenophaga* **b** *Methylobacter* and *Methylotenera* and **c** *Sulfuricurvum* and *Thiobacillus*. Oxygen concentration is shown on pseudo-log$_{10}$ scale, to include samples with no oxygen (zero). Boxplots summarize relative abundances of **d** hydrogenotrophs, **e** methylotrophs, **f** thiotrophs, and **g** oxygen concentrations based on water age categories (upper and lower quartiles and whiskers (each representing 25% of the data), median (line) and outliers (dots)). Relative sequence abundances of the aerobic/denitrifying clades tend to peak at hypoxic conditions (~0.5 mg L$^{-1}$) potentially because these waters contain both sufficient electron donors (hydrogen, methane, sulfur) and electron acceptors (oxygen, nitrate). *n* is given in **f** and applies to all panels.

Sulfur-oxidizing bacteria including *Sulfuricurvum* sp., *Thiobacillus* sp., *Thiomicrorhabdu*s, and *Sulfurimonas* were also widespread and abundant in the studied aquifers (Fig. 6c, Supplementary Fig. 10). *Thiobacillus* (MAG-20) was able to oxidize sulfide and thiosulfate to sulfate using oxygen, but also had a complete nitrate reduction pathway (Fig. 7). A similar lifestyle was likely for PFJX01 within the *Thiobacillaceae* (MAG-21), *Sulfurimicrobium* (MAG-22), *Rhodoferax* (MAG-23, −24), UBA2250 (MAG-26, −27) and ETT8 (MAG-29) affiliating with *Rhodocyclaceae* and *Rhodobacteraceae*. Some members of these clades were shown to be active facultatively anaerobic denitrifying sulfur oxidizers in the terrestrial subsurface[49,50]. The versatile sulfur metabolisms of these clades could explain previous observations of pyrite oxidation[14], as well as the presence of sulfur reducers like *Desulfuromonas* which can reduce elemental sulfur[51] and replenish the sulfide pool.

We found obligate aerobic ammonium oxidizing *Nitrosomonas*, its MAG encoded a partial *pmoA/amoA* and a *hao* to oxidize ammonium to nitric oxide (Supplementary Data 5). Like the activation of methane, the ammonium monooxygenase requires molecular oxygen. The biomass produced by the diverse, coexisting autotrophs in turn may support multitudes of archaeal and bacterial heterotrophs, including DPANN archaea and *Patescibacteria*. Numerous MAGs represented *Flavobacteriales*, *Cytophagales*, *Chitinophagales,* and *Bacteroidales*, heterotrophs with an appetite for oligosaccharides and the necessary transport systems. In the archaeal community aerobic *Woesearchaeales* were particularly diverse and abundant in all aquifers except those with old groundwaters, and in the bacterial community potentially heterotrophic *Comamonadaceae* and *Pseudomonadaceae* were also very abundant (Supplementary Fig. 14b, c). Heterotrophic clades remineralize organic carbon compounds to carbon dioxide, which could then be recycled by hydrogenotrophic methanogens and other autotrophs to produce biomass.

### Dark oxygen is an electron acceptor in aquifers

As water infiltrates through recharge zones and flows through the aquifers, the dissolved oxygen (DO; assumed to be at saturation equilibrium initially) is consumed by microbial respiration, decomposition of organic matter, or by reacting with reduced minerals[52]. These oxygen-consuming reactions often reduce the DO content of groundwater to below detection limits, particularly in groundwater with long residence times that has been out of contact with the atmosphere for many years, centuries, or even millennia[52,53]. To our surprise we detected low concentrations of DO in most groundwater samples including those of deeper aquifers containing geochemically mature groundwater (Fig. 9a). Using the known stoichiometry of relevant microbial metabolisms, we estimated that the consumed oxygen (DO saturation at infiltration minus measured DO concentration in the obtained samples) could sustain more microbial cells than we observed in 85% of aquifers (59 of 70; Supplementary Data 8). In the remaining 11 aquifers, the DO content was below the detection limit. The occurrence of more than 0.3 mg L$^{-1}$ DO in many old groundwaters of deeper and confined aquifers was remarkable and suggests that oxygen is present in ecosystems that are often assumed to be anoxic[52,54].

To investigate whether oxygen may have been introduced during sampling, we carried out oxygen isotope analyses of molecular oxygen (O$_2$). Indeed, some oxygen isotope data for DO were consistent with groundwater that is in equilibrium with atmospheric oxygen ($\delta^{18}O_{O_2}$ = ~+23.9‰ ± 0.1‰, Fig. 9b), indicating air contamination during sampling, or the presence of atmospheric-derived oxygen in the aquifers. However, at certain sites (GW218, GW265) we observed markedly lower $\delta^{18}O_{O_2}$ values (as low as +21‰), while also finding elevated O$_2$:Ar ratios (Fig. 9b). The lower $\delta^{18}O_{O_2}$ and higher O$_2$:Ar ratios interestingly fell along a trend, representing the simulated addition of DO with a $\delta^{18}O_{O_2}$ value much lower than that of air-equilibrated water

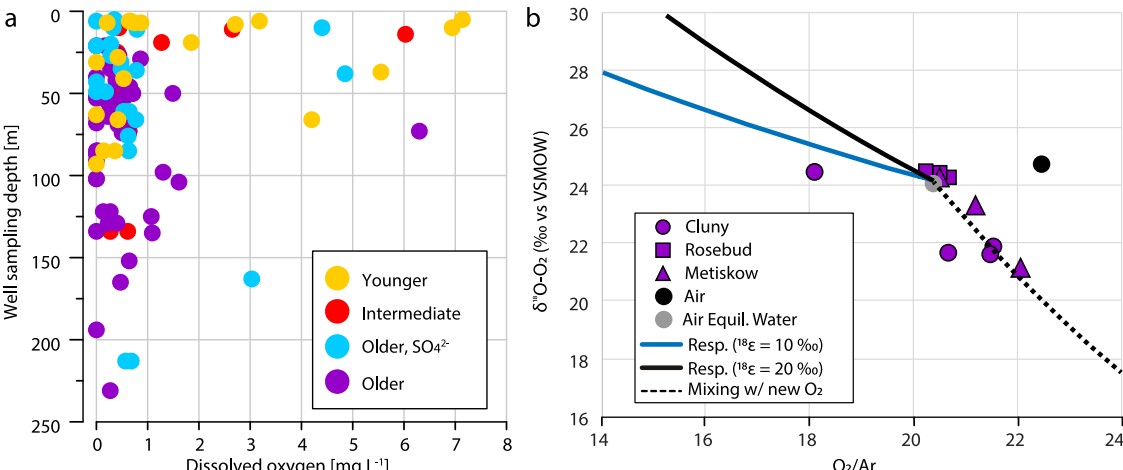

**Fig. 9 | Concentration and isotopic signature of oxygen in groundwater samples. a** Depth profile and dissolved oxygen concentration (mg L$^{-1}$) in the groundwater samples. **b** $^{18}$O-O$_2$ isotopic signature over oxygen:argon ratio (O$_2$/Ar). The composition of lab air-equilibrated water and lab air are represented by a gray and black circle, respectively. Error bars are 1 standard deviation (based on repeat analysis of lab air, $n = 13$) and are smaller than the symbols. The blue and black solid lines show different isotope effects that would be caused by fractionation (e) due to respiration (resp.). During consumption of O$_2$ by respiration (decreasing O$_2$:Ar ratio), there is a preferential accumulation of $^{18}$O in the remaining O$_2$ pool−leading to higher $\delta^{18}$O$_{O_2}$ values. As the degree of this isotope fractionation can vary the two solid lines represents two hypothetical but realistic scenarios for how the O$_2$ pool might change with microbial respiration. The black dashed line represents a mixing line between air-equilibrated water mixed with a hypothetical 'new source of O$_2$' that has a very low $\delta^{18}$O$_{O_2}$ (−20‰ vs V-SMOW). Such isotopically light oxygen is consistent with the biological formation of O$_2$. *V-SMOW* Vienna Standard Mean Ocean Water.

(dashed trend line in Fig. 9b). The variation among the triplicate samples may be explained by the variable presence of atmospheric-derived oxygen and/or by varying microbial respiration, which will decrease the O$_2$:Ar and increase $\delta^{18}$O$_{O_2}$ values. In fact, mixing and biological processes are likely both at work. Most remarkably, however, there is no plausible scenario to our knowledge whereby air contamination or microbial respiration could lead to the observed increasing trend in O$_2$:Ar ratio coupled with decreasing $\delta^{18}$O$_{O_2}$ values. That is, air contamination would bring O$_2$:Ar ratios and $\delta^{18}$O$_{O_2}$ back toward air-equilibrated values of +23.9‰[55] while microbial consumption of oxygen would lower O$_2$:Ar while increasing $\delta^{18}$O$_{O_2}$[56]. The most parsimonious explanation for the observed trend would be the in-situ production of O$_2$ with a very low $\delta^{18}$O$_{O_2}$ (e.g., −20‰). Given the low $\delta^{18}$O (of H$_2$O) of groundwaters in Alberta (−18.6‰ ± −2.0‰; mean ± SD; $n = 144$, Supplementary Data 1), which would be the source of oxygen for any in-situ production of O$_2$, even a small amount of production would have a large impact on lowering the overall $\delta^{18}$O$_{O_2}$ in groundwater, in the direction of our observations. In the absence of light, oxygen can be produced by water radiolysis[57], or microbially by chlorite dismutation[58], nitric oxide dismutation[28,59], and H$_2$O$_2$ dismutation[60].

## Microbial production of dark oxygen via dismutation

Microbes can produce O$_2$ via chlorite dismutation, a process carried out among others by microbes of the genera *Dechloromonas*[58], *Dechlorobacter*, *Dechlorosoma*, *Azospira*, *Azospirillum*[61], *Nitrospina* and *Nitrobacter*, all of which were present in the GOWN aquifers based on the 16S rRNA gene analyses. ASVs affiliating with the genus *Dechloromonas* were among the most abundant in the study accounting for up to 30% of all bacterial sequences in some samples. We found 13 chlorite dismutase genes mainly from *Dechloromonas*, *Nitrospina*, *Nakamurella*, and *Magnetospirillum* species in old, anoxic/hypoxic (DO: 0−0.55 mg L$^{-1}$) groundwaters of GW144, GW218, GW265, and GW972 (Supplementary Data 9). The chlorite dismutase genes were detected at considerable sequencing depths ranging from 4 to 212×, indicating a moderate to high abundance of the organisms having this gene. Particularly two chlorite dismutases assigned to *Magnetospirillum* had very high sequencing depth (207× and 212×). Groundwater can also contain a high diversity of nitric oxide dismutation genes[62] and

microbes affiliating with *Methylomirabilaceae* that can potentially dismutate nitric oxide were present in many samples. We also found nitric oxide dismutase genes (*nod*) in high-quality MAGs of *Sediminibacterium* (MAG-30) and *Algoriphagus* (MAG-31; Fig. 7; Supplementary Data 9). Both lineages were recently shown to have a *nod*[63]. The *nod* affiliating with *Sediminibacterium* was sequenced at a depth of 700× corroborating the very high abundance of *Sediminibacterium* in the community (12%). Nitric oxide reductase is related to *nod* but it can be identified and distinguished by several diagnostic amino acid residues in the active center of the enzyme (Supplementary Fig. 15). It was recently shown that nitric oxide dismutating *Nitrosopumilus* sp. can leak intracellularly produced oxygen into the medium possibly supporting other aerobic organisms[59]. It was also reported that even *Pseudomonas aeruginosa* seem to be able to dismutate nitric oxide and release peaks of up to 20 μM oxygen into their surroundings[64]. Lineages affiliating with both, *Nitrosopumilus* and *Pseudomonas* were widespread and abundant in the studied groundwaters. Abundant membrane-bound dismutases in the studied aquifers and their release of oxygen into the surroundings could explain the observed presence of $^{18}$O-depleted oxygen in old groundwaters.

## Microbial lineages, metabolic capabilities, and niches in groundwater ecosystems

Our findings suggest that hydrogen is a basal energy source fueling a rich mosaic of microbial metabolisms. The extensive microbial productivity in the studied groundwater ecosystems (Fig. 10) underlines the importance of hydrogen in the terrestrial subsurface[65]. Hydrogenotrophic methanogens produce methane, which in turn explains the high abundance of methanotrophs and widespread genes involved in methanotrophy. Substantial aerobic and anaerobic methane oxidation can reduce greenhouse gas emissions from aquifers containing coal beds and shales with relevance for carbon budgets and climate change. Hydrogen also appears to drive extensive microbial sulfur cycling. Sulfate reducers produce sulfide which is then oxidized by thiotrophic microbes using either oxygen or nitrate as an electron acceptor in their metabolic pathways. The microbial community composition, the detected metabolic capabilities, and aqueous and isotope geochemistry are all aligned and suggest that groundwater

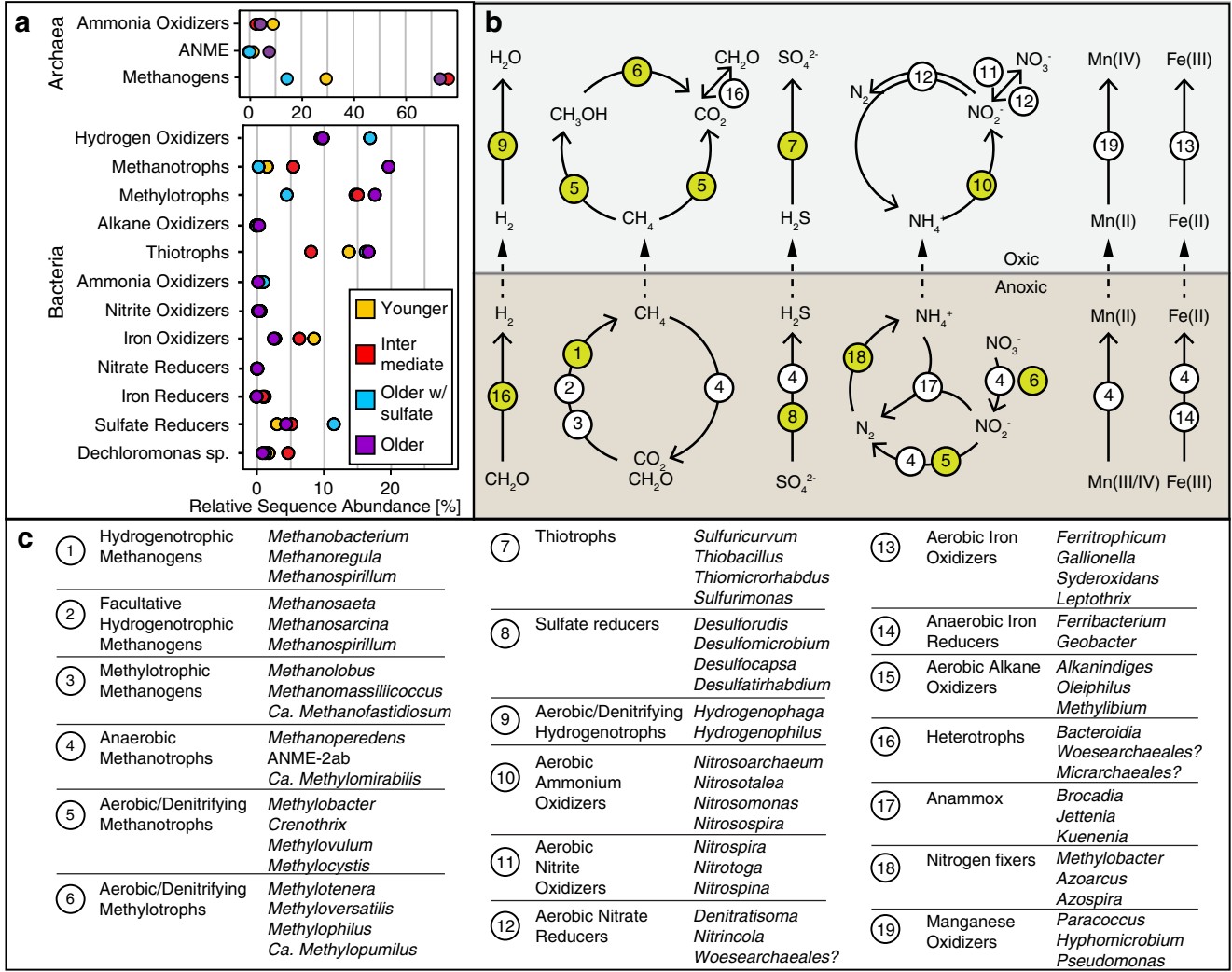

**Fig. 10 | Microbial guilds and potential community functions. a** Relative sequence abundances of microbial guilds. *ANME*: Anaerobic methane-oxidizing archaea **b** Schematic overview of element cycling inferred from microbiological and geochemical analyses. Numbers refer to key microbial genera and their potential function listed in **c**. $CH_2O$: biomass, $H_2$: hydrogen, $H_2O$: water, $CO_2$: carbon dioxide, $CH_4$: methane, $CH_3OH$: methanol, $SO_4^{2-}$: sulfate, $H_2S$: sulfide, $NO_3^-$: nitrate, $NO_2^-$: nitrite, $N_2$: nitrogen, $NH_4^+$: ammonium, Mn: manganese, Fe: iron.

**c** Microbial lineages and their potential function. The list of microbes is not exhaustive and mainly contains the most abundant genera observed in this study. Processes in **b** depicted with white circles are supported by geochemistry and/or metabarcoding, and processes depicted by lime-green circles are additionally supported by the respective metabolic capabilities present in metagenome-assembled genomes.

microbial communities utilize the entire breadth of known biochemically available redox potential, from hydrogen to oxygen. Unlike in many other aquifers in which cell numbers decrease with increasing groundwater age, cell numbers increased with age in groundwater obtained from aquifers in Alberta, corroborating that these aquifer communities are fueled by autochthonous energy sources derived from within the subsurface. Considering the size of the investigated area we conclude that global subsurface biomass may be underestimated, particularly in aquifers containing organic carbon-rich strata, such as coal beds and shales. The geochemical and microbiological data suggest that oxygen is an important electron acceptor for microbial metabolisms in the studied aquifers. The presence of aerobic microbes in deep aquifers and bedrock has been previously reported[66], yet the mechanisms of deep oxygen migration or in situ production remain unclear. While co-migration of some oxygen with the aging groundwater could theoretically explain the observed microbial productivity, the presented oxygen isotope analyses, and the abundance of microbial dismutase genes capable of producing oxygen suggest that at least some portion of the oxygen is generated in-situ. Most of the hypoxic waters surveyed here are several thousand

to >10,000 years old based on tritium and $^{14}C$ data, supporting that even deep subsurface ecosystems provide niches for aerobic microorganisms[67]. The production of dark oxygen that we postulate in this work could provide a mechanism for previously reported isotopically light groundwater oxygen anomalies that have lacked a clear explanation thus far[52,54,68–71]. Microbial dark oxygen production in subsurface ecosystems may thus be relevant for the functioning and evolution of the geobiosphere, as it provides a source of oxygen independent of light, on Earth as well as potentially on other celestial bodies.

## Methods

### Groundwater wells

Approximately 40–90 groundwater wells are sampled annually for water quality by Alberta Environment and Protected Areas[8] and analyzed for a variety of geochemical, isotopic, and microbial parameters. For this study, we analyzed 138 groundwater samples from 95 monitoring wells, sampled from January 2016 to July 2020. 38 wells are screened within Neogene-Quaternary surficial deposits while 57 wells reach sedimentary bedrock, usually of Paleogene or Cretaceous age

(Supplementary Data 1). Well depths in this study ranged from five to 231 m (mean = 60 m). The sedimentary bedrocks in which groundwater monitoring wells are completed include the Paskapoo and equivalent ($n = 25$), Horseshoe Canyon and equivalent ($n = 13$), Belly River Group ($n = 9$), Bearpaw ($n = 4$), Milk River ($n = 3$) and Loon River ($n = 2$) formations. One well was not assigned with a geological formation. For further details concerning the location and geology refer to the SI.

## Sampling

All wells were sampled at least once, while select wells were sampled during different seasons (up to three), or repeatedly to obtain sampling replicates (up to three). Some wells were installed as clusters, with multiple wells (up to three) at the same location completed at different depths within the same formation or across multiple formations. All wells were purged to improve sample quality[72] and samples were collected after field parameters (dissolved oxygen - DO, oxidation-reduction potential - ORP, temperature, electrical conductivity - EC) had stabilized, indicating representative aquifer water. All wells were sampled following the same procedure and the samples were stored at 4 °C until further processing. Dependent on the location of the wells, the samples were in transit at 4 °C between 1 and 7 days. Tritium ($^3$H), and carbon-14 of dissolved inorganic carbon ($^{14}C_{DIC}$) samples were collected with a submersible pump (SP). Samples for other water quality parameters, isotopes, dissolved gas, and microbiological samples were collected with a bladder pump (BP). The SP is placed 2 m below the expected drawdown level for purging and sample collection. The submersible pump has polyethylene tubing that does not get cleaned, and Nalgene tubing that has been cleaned with a general laboratory detergent (Sparclean) followed by a rinse with ultrapure (18 MΩ) water. The BP is either placed in the vicinity of the SP or just above the screened interval for sample collection and has acid-washed, methanol, and ultrapure water-rinsed Teflon tubing. For microbiological analyses we collected 1000 mL$^{-1}$ groundwater in a sterile Nalgene bottle. The sample was sent to the laboratory at 4 °C and used for microbial cell cryo-preservation, microbial cell counts, and nucleic acid extractions.

## Dissolved oxygen measurements

Dissolved oxygen measurements were conducted with multiparameter continuous water quality monitoring sondes, which have lifetime-based optical luminescence sensors with a resolution of 0.01 mg L$^{-1}$ and an accuracy of ±0.1 mg L$^{-1}$. The specific sondes and sensors used during the 2016–2017 sampling campaigns were a YSI 6600 XLM (Dissolved Oxygen – ROX; YSI, Yellow Springs, USA), EXO1 or EXO2 (Optical Dissolved Oxygen Smart Sensor; YSI, Yellow Springs, USA), In-Situ AquaTROLL 600 (Rugged Dissolved Oxygen Sensor with RDO-X cap; In-situ, Fort Collins, USA), or an Ott Hydrolab DS5X (Hach LDO; Hach, Düsseldorf, Germany). The sample tubing was connected to a ~400–900 mL flow through cell and field parameters were logged at 1–2 min intervals such that a full volume of water was displaced in between readings, using the default or accelerated averaging modes (5–40 second filtering), until parameter stabilization (±0.2–0.3 mg L$^{-1}$) was reached for 15 consecutive readings.

## Dissolved oxygen sensor calibration

Before each field trip, calibrations of the dissolved oxygen sensors were performed in the lab. Calibrations for water-saturated air were conducted according to the user manuals of the probes YSI EXO or 6-Series (YSI, Yellow Springs, USA) and the Aqua TROLL 600 (In-situ, Fort Collins, USA) or air-saturated water according to the user manual of the Hydrolab DSX6, DS6 and MS6 (Hach, Düsseldorf, Germany). For water-saturated air, the calibration cup was filled with 1/8 to ½ of an inch of tap water. A clean and dry dissolved oxygen sensor was inspected for damages and cracks, then covered with a clean calibration guard. The sensor and guard were inserted into the calibration cup and left loosely affixed to the probe to allow for venting. The setup was left to rest for 2–15 min out of direct sunlight to allow the temperature and pressure to equilibrate with water-saturated air. For air-saturated water, 1 L of room temperature water that had been at equilibrium with atmospheric pressure for at least 12 h, was vigorously shaken for one minute, poured into the calibration cup with the Hach LDO sensor cap (Hach, Düsseldorf, Germany) and the temperature sensor fully submersed. The calibration cup cap was lightly placed over the calibration cup to allow for air exchange and kept out of direct sunlight for 3–5 min. Barometric pressure was measured with a barometer in an EXO Handheld (YSI, Yellow Springs, USA) or Vaisala PTB220 Digital Barometer (Vaisala Oyj, Vantaa, Finland) was recorded on the calibration record, and entered to the calibration software in mmHg. A one-point calibration was conducted for water-saturated air or air-saturated water. Readings were observed until they stabilized for 40 seconds, the current value was recorded in the calibration record, and the calibration point was accepted. Percent saturation was converted to mg L$^{-1}$ using measured temperature (thermistor) and salinity (conductivity sensor) and formulas from Standard Methods for the Examination of Water and Wastewater[73]. The health of the sensor cap was tracked by monitoring optical dissolved oxygen gain within 0.7 to 1.4 and replaced every ~2 years.

## Chemical analyses

Groundwater samples for major cations and anions were collected in polyethylene bottles and analyzed at ALS Global by Inductively Coupled Plasma-Mass Spectrometry (ICP-MS) and Ion Chromatography, respectively. Total and phenolphthalein alkalinity was determined by titration, and bicarbonate and carbonate concentrations were subsequently calculated. Dissolved organic carbon was measured by high-temperature combustion on a total organic carbon analyzer with infrared detection. Dissolved trace element samples were filtered in the field using 0.45 μm in-line capsule filters, preserved to pH <2 with nitric acid, and analyzed at InnoTech Alberta by ICP-MS.

## Gas analyses

Dissolved gas samples were collected with a bladder pump. Evacuated crimp-top glass serum bottles with butyl septa and preserved with mercuric chloride were filled by piercing the septa with a needle. Gas composition was determined in the Applied Geochemistry Laboratory at the University of Calgary by gas chromatography. The gas dryness parameter is defined as the ratio between the concentrations of methane and those of higher $n$-alkanes. The isotopic composition of methane and carbon dioxide was analyzed in the Isotope Science Laboratory at the University of Calgary on a ThermoFischer MAT 253 isotope ratio mass spectrometer (IRMS) coupled with a Trace GC Ultra and GC Isolink (ThermoFisher) and reported relative to V-PDB for $\delta^{13}C$ and V-SMOW for $\delta^2H$. The precision for carbon isotope analyses was higher than ±0.5‰ for hydrocarbons, ±0.2‰ for carbon dioxide, and ±2‰ for $\delta^2H$ of methane.

## Groundwater dating

All samples for $^{14}C_{DIC}$ analyses were shipped to the A.E. Lalonde Laboratory in Ottawa for analysis by accelerator mass spectrometry (AMS). Sample pretreatment techniques can be found in ref. 74. Radiocarbon analyses are performed on a 3MV tandem accelerator mass spectrometer built by High Voltage Engineering (HVE). We used the fractionation-corrected fraction modern (the $F^{14}C$ value) for post-bomb $^{14}C$ data according to the amended conventions[75]. Tritium ($^3$H) is a radioactive isotope of hydrogen with a half-life of 12.4 years. Tritium concentrations are measured in tritium units (TU) where 1 TU is defined as the presence of one tritium in $10^{18}$ atoms of hydrogen. For samples from the 2016–2017 sampling campaign, tritium in the obtained groundwater samples was analyzed at the A.E. Lalonde

Laboratory in Ottawa by electrolytic enrichment and the standard method of liquid scintillation counting with a precision of 0.8 TU.

## Carbon, oxygen, nitrogen, and sulfur isotope analyses

Most stable isotope analyses were performed in the Isotope Science Laboratory at the University of Calgary (https://www.ucalgary.ca/labs/isotope-science-lab/techniques). Dissolved inorganic carbon (DIC) isotope samples were filtered in the field using an in-line 0.2 μm capsule filter and collected in a 125 mL clear glass flint bottle with a cone cap. $CO_2$ was produced from DIC by acid attack and $\delta^{13}C$ was analyzed as described in the gas analyses section. For the isotopic composition of sulfate, samples were filtered in the field with an in-line 0.2 μm capsule filter and collected in 1 L wide mouth HDPE plastic bottles. Dissolved sulfate was converted into barium sulfate and analyzed using a ThermoQuest Finnigan Delta Plus XL IRMS coupled with a Fisons NA 1500 Elemental analyzer for $\delta^{34}S_{SO4}$ and a HEKAtech HT Oxygen Analyzer with a Zero Blank autosampler for $\delta^{18}O_{SO4}$. $\delta^{34}S_{SO4}$ is reported relative to V-CDT and $\delta^{18}O_{SO4}$ is reported relative to V-SMOW. Precisions for both $\delta^{18}O_{SO4}$ and $\delta^{34}S_{SO4}$ is ±0.5‰. The isotopic composition of nitrate was determined on $N_2O$ generated by the denitrifier technique using a Thermo Scientific Delta V Plus IRMS coupled with a Finnigan MAT PreCon. Precisions for $\delta^{15}N_{NO3}$ and $\delta^{18}O_{NO3}$ are ±0.3‰ and ±0.7‰, respectively.

## Oxygen isotope analyses on dissolved oxygen

Samples for the analysis of oxygen isotope ratios of dissolved $O_2$ and $O_2$/Ar ratio measurements in groundwater from monitoring wells were filled directly into 20 mL headspace vials allowing many volumes of overflow to displace air, poisoned with 100 μL of a saturated zinc chloride solution, quickly crimp-sealed with butyl septa (with no headspace bubble). At the laboratory, a 5 mL headspace was established by replacement with ultra-high purity helium. Vials were equilibrated with headspace on a shaker table at room temperature for several days. Aliquots of the headspace (containing $O_2$ and Ar from the water sample) were directly injected into the sample inlet system (including a 2 m, 5 Å molecular sieve GC column) and routed into the isotope ratio mass spectrometer (Isoprime 100, multi-collector). Injections of lab air and lab air-equilibrated water were used as working standards. Based on repeat analyses the precision of the $\delta^{18}O$ measurements was ±0.1‰ and of the $O_2$/Ar was ±0.05.

## Cell staining, enumeration, and biomass yield estimations

To fix cells we added 4 mL formaldehyde solution (37%) to a 50 mL groundwater sample (f. c. -2.7%). The sample was stored at 4 °C until further processing. 1 mL sample was diluted in 10 mL 1× phosphate buffered saline (PBS) and filtered onto a polycarbonate filter (0.1 μm pore size, 25 mm diameter, Millipore Sigma) using a mixed cellulose ester membrane filter (0.45 μm pore size, Sartorius) as support. The filter was rinsed with 10 mL of 1× PBS, dried, and stored until further processing. A section of each filter was stained with 1 μg mL$^{-1}$ DAPI (4',6-diamidino-2-phenylindole) for 10 min at room temperature, washed with deionized water and 80% ethanol, and dried. Filter pieces were mounted on microscope slides using 4:1 Citifluor:Vectashield solution (VWR and Vector Laboratories) and stored at −20 °C. Cells were visualized and counted using an Axio Imager A2 (Zeiss, Jena, Germany) equipped with an X-Cite 120 LED (Excelitas, Waltham, USA) fluorescence light source and 12.5 × 12.5 mm ocular grid. -40 grids per filter were counted to obtain a robust dataset. To estimate the cell number that can be sustained per mol oxygen (Supplementary Data 8) we used published values for microbial cell content[76] and biomass yield[77].

## 16S rRNA gene library preparation and amplicon sequencing

100–400 mL groundwater sample was centrifuged at 4000 × $g$ for 1 h at 4 °C. The supernatant was discarded, the pellets combined in a 2 mL tube and stored at −80 °C until further processing. (Note: We test collected cells of 8 wells also via filtration on 0.1 μm pore size GTTP filters (Millipore Sigma), sequenced DNA, and compared the community structures to those derived of the pellets. The communities were nearly identical). Genomic DNA was extracted using the DNeasy PowerLyzer PowerSoil Kit (12855-100, QIAGEN) according to manufacturer's protocol with a minor modification; cells were lysed by bead beating at 4 m s$^{-1}$ for 45 s using a Bead Ruptor 24 (OMNI). Extraction blanks were processed to detect potential laboratory contamination during extraction. DNA concentrations were measured fluorometrically using a Qubit 2.0 (Thermo Fisher Scientific, Canada). The bacterial 16S rRNA gene v3-4 region was amplified using S-D-Bact-0341-a-S-17 (5'-CCTACGGGAGGCAGCAG-3') and S-D-Bact-0785-a-A-19 (5'-GACTACHVGGGTATCTAATCC-3')[78]. The archaeal 16S rRNA gene v6-9 region was amplified using the primer pair S-D-Arch-0915-a-S-20/S-*-Univ-1392-a-A-15 (5'-AGGAATTGGCGGGGGAGCAC-3', 5'-ACGGGCGG TGTGTRC-3')[78]. PCRs consisted of 8 μL (1–10 ng) DNA template, 2.5 μL of each primer (f.c. 1 μM), 12.5 μL 2× Kapa HiFi HotStart Ready Mix (Kapa Biosystems, Wilmington, MA, USA) and PCR-grade water ad 25 μL. For bacteria, a touchdown PCR program was used for improved annealing: initial denaturation at 95 °C for 3 min, 10 cycles of 95 °C for 30 sec, 60 °C for 45 sec (touchdown −1 °C per cycle), 72 °C for 60 sec, followed by 20 cycles of 95 °C for 30 sec, 55 °C for 45 sec, 72 °C for 60 sec, and a final extension at 72 °C for 5 min. For archaea, the touchdown PCR program was: initial denaturation at 95 °C for 5 min, 10 cycles of 95 °C for 30 sec, 62 °C for 45 sec (touchdown −1 °C per cycle), 72 °C for 60 sec, followed by 20 cycles of 95 °C for 30 sec, 60 °C for 45 sec, 72 °C for 60 sec, and a final extension at 72 °C for 5 min. PCRs were performed in triplicate, pooled, and purified using 56 μL Agencourt AMPure XP beads (Beckman Coulter, Indianapolis, USA) per pooled PCR product (-65–75 μL) following manufacturer's instructions. Amplicons were indexed using 5 μL purified PCR product, 5 μM of each Index Primer (f.c. 1 μM), 25 μL 2× Kapa HiFi HotStart Ready Mix and 10 μL PCR-grade water. Indexing PCR program: initial denaturation at 95 °C for 3 min, 10 cycles of 95 °C for 30 sec, 55 °C for 45 sec, 72 °C for 60 sec, and a final extension at 72 °C for 5 min. Indexed amplicons were purified with Agencourt AMPure XP beads. The concentration and size of indexed amplicons were checked with a Qubit 2.0 fluorometer and Agilent 2100 Bioanalyzer system (Agilent Technologies, Mississauga, ON, Canada), respectively. Indexed amplicons were pooled in equimolar amounts and sequenced using Illumina's v3 600-cycle (paired-end, MS-102-3003) reagent kit on an Illumina MiSeq benchtop sequencer (Illumina Inc., San Diego, CA, USA) after all DNA extraction blanks and PCR reagent blanks were confirmed for negative amplification.

## Community analyses

Raw amplicon sequences were analyzed using *DADA2* v1.16[79]. Briefly, forward, and reverse reads were quality-trimmed to 275 bp and 205 bp, respectively, and primer sequences (17 bp forward, 21 bp reverse) were removed. Reads with more than two expected errors were discarded, paired reads were merged, and chimeric sequences were removed. Species level taxonomy was assigned with the silva_nr_v138_train_set and silva_species_assignment_v138. After quality control and the removal of blanks and technical replicates we obtained 64 archaeal and 110 bacterial amplicon datasets. Archaeal datasets contained a total of $3.89 \times 10^5$ sequence reads belonging to 2633 unique ASVs. Archaeal samples had on average $6083 \pm 6387$ reads (mean ± standard deviation) and $69 \pm 81$ ASVs (Supplementary Data 10). Bacterial datasets comprised a total of $4.65 \times 10^6$ sequence reads belonging to 14665 unique ASVs. Bacterial samples had on average $4.23 \pm 1.57 \times 10^4$ reads and $272 \pm 221$ unique ASVs (Supplementary Data 11). Supplementary Data 10 and 11 also include NCBI Project and Sample Accession numbers for each archaeal and bacterial dataset. Archaeal and bacterial ASV nucleotide sequences and

taxonomy are listed in Supplementary Data 12 and 13, respectively. The archaeal or bacterial ASV-by-sample table (Supplementary Data 3, 4) was used to determine the number of observed ASVs, absolute singletons, relative singletons, relative abundance, and composition. Alpha diversity (richness, Shannon entropy, Inverse Simpson Diversity, and Chao1 estimated richness) was calculated from the ASV-by-sample table using a subsampling approach to account for unequal sampling effort. We used 1008 and 6796 randomly chosen reads from each archaeal and bacterial sample, respectively. Differences in diversity between conditions were tested using the Wilcoxon signed rank-test (*ggsignif*) as implemented in *ggplot2*[80]. Bray-Curtis dissimilarities between all samples were calculated and used for two-dimensional nonmetric multidimensional scaling (NMDS) ordinations with 20 random starts. The further two samples (circles) are apart in the NMDS the more different are their underlying communities. Environmental parameters were tested with a partial Redundancy Analysis using Hellinger-transformed ASV data. All analyses were carried out with VisuaR v02 (https://github.com/EmilRuff/VisuaR) a workflow based on the R statistical environment v4.1.0-v4.2.1, including among others the packages *vegan*[81], *ggplot2*[80] as well as custom R scripts. The VisuaR workflow, the used packages, and versions are available in Supplementary Data 14.

### Metagenome sequencing and analysis

Metagenomes were processed and sequenced at the Center for Health Genomics and Informatics in the Cumming School of Medicine, University of Calgary. Genomic DNA was sheared into ~350 bp fragments using a S2 focused-ultrasonicator (Covaris, Woburn, MA). Libraries were prepared using the NEBNext Ultra II DNA Library Prep Kit for Illumina (E7103, New England Biolabs, Ipswich, MA) according to the manufacturer's protocol, including size selection with SPRIselect magnetic beads (B23318, Beckman Coulter, Indianapolis, IN) and PCR enrichment (eight cycles) with NEBNext Multiplex Oligos for Illumina (E7335L, New England Biolabs, Ipswich, MA). DNA concentrations were estimated using qPCR and the Kapa Library Quantitation Assay for Illumina (Kapa Biosystems, Wilmington, MA). Genomic DNA was sequenced on an Illumina NovaSeq 600 sequencer (Illumina, San Diego, CA) using a 300 cycle (2 × 150 bp) S1 flow cell. Quality control was performed on raw, paired-end Illumina reads using BBDuk v38.90[82], including trimming, filtration of contaminants, and clipping low-quality ends. Reads that passed quality control were assembled into contigs using Megahit v1.2.9[83] and contigs of <500 bp were not included in subsequent steps. Reads from each sample were mapped to each of the 5 samples using BBMap v38.90[82] and sequencing depth profiles were generated using the 'jgi_summarize_bam_contig_depths' function from MetaBat2 v2.15[84]. Assembled contigs were binned into metagenome-assembled-genomes (MAGs) using MetaBat2, CONCOCT v1.1.0[85] and MaxBin2 v2.2.7[86]. DAS-Tool v1.1.3[87] was used to integrate MAGs produced by the three binning tools. The MAG properties including completeness and contamination were assessed by CheckM v1.2.0[88]. To estimate the relative abundance of MAGs we determined the percentage of reads that mapped to each bin and the percentage that mapped to unbinned contigs using *checkm coverage* and *checkm profile* and then calculated (% reads mapped to bin) * (100 − (% reads mapped to unbinned contigs)). MAGs were classified using GTDB-tk (version 2.1.0, database release r207)[89]. Metagenomic short reads were mapped to the SILVA SSU reference database v138[90] to assign nearest taxonomic units, as well as full-length 16S/18S rRNA gene sequences were reconstructed from metagenomes using phyloFlash v3.4[91] and compared to ASVs using blastn v2.12.0[92]. Transfer RNA, ribosomal RNA, CRISPR elements, and protein-coding genes including nitric oxide dismutase (*nod*) and chlorite dismutase (*cld*) coding genes were predicted and annotated using MetaErg v2.2.x[93]. Amino acid sequences on contigs were additionally searched against representative *nod*

and *cld* sequences from NCBI using blastp v2.12.0 (e-value 1e^{−10}) and aligned using Clustal Omega v1.2.4[94].

### Contamination control

To avoid and monitor contamination we have included controls at each step of the experiments. We have sterilized the sampling gear after each well, we included duplicate blind controls of regular water samples by using encrypted sample names, and we included blank controls for cell counting and sequencing. We have no indication that sample handling or DNA extraction introduced organisms that are known contaminants of low biomass samples[95] or DNA extraction kits[96]. The storage in Nalgene bottles which could have had a potential enrichment effect, e.g., promoting growth of aerobic organisms after sampling, had no effect on community structure (Supplementary Fig. 11), as alpha diversity, beta diversity and composition did not show any differences when grouped into categories reflecting storage duration.

### Reporting summary

Further information on research design is available in the Nature Portfolio Reporting Summary linked to this article.

## Data availability

The archaeal and bacterial 16S rRNA amplicon data generated in this study have been deposited in the NCBI SRA archive under BioProject accession number PRJNA861683. The shotgun metagenomic data and metagenome-assembled genomes (MAGs) have been deposited in the SRA archive under BioProject accession number PRJNA700657. The comprehensive environmental data generated in this study have been deposited in the PANGAEA archive under accession number 952473[97].

## Code availability

The most recent version of the bioinformatic workflow VisuaR v02 used to perform 16S rRNA gene-based community analyses is publicly available at Github (https://github.com/EmilRuff/VisuaR), the exact VisuaR analyses featured in this paper are deposited in Supplementary Data 14.

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

## Acknowledgements
We are very grateful to Joanna Borecki, James Rogans, Dennis Rollag, Vien Lam, and other scientists and staff of the Groundwater Observation Well Network operated by Alberta Environment and Protected Areas (https://www.alberta.ca/groundwater-observation-well-network.aspx) for providing access to groundwater monitoring wells, for sampling and providing highest quality groundwater samples, and for sharing measurement results and expertise. We greatly appreciate the support and expertise of Carmen Li regarding nucleic acid sequencing. We thank Manuel Kleiner and Xiaoli Dong for support and discussions regarding bioinformatics, and Ranjani Murali and Claire Elbon for discussions concerning oxidases and dismutases. We thank Anirban Chakraborty for insights on sample handling, Steven Taylor and Veith Becker for support with isotopic measurements and interpretation, and Rachel HR Stanley for help with oxygen isotope mass spectrometry. This work was supported by a grant from the Simons Foundation (824763, S.E.R.), an Alberta Innovates Technology Futures (AITF)/Eyes High Postdoctoral Fellowship (S.E.R.), start-up funds by the Marine Biological Laboratory, Woods Hole (S.E.R.), by Alberta Innovates Energy and Environment Solution (AIEES)—Project: Geochemical resource characterization of Alberta groundwater (B.M.), and Alberta Innovates Water Innovation Program (AI-WIP)—Project: Occurrence, origin and the fate of aqueous contaminants in Alberta groundwater (B.M.), and by the Canada Research Chairs Program (CRC-2020-00257, M.S.).

## Author contributions
S.E.R., B.M., and M.S. conceptualized the study. S.E.R., P.H., B.M., and M.S. secured funding. S.E.R., P.H., C.N.M., I.H.d.A., S.D.W., A.S., and M.S. devised methodology. S.E.R., P.H., I.H.d.A., M.N., M.D., S.C., L.C., O.O.K., A.S., S.B., and S.D.W. processed samples. S.E.R., P.H., I.H.d.A., C.N.M., M.D., and S.D.W. carried out analyses. S.E.R., P.H., I.H.d.A., S.D.W. visualized data. S.E.R. and M.S. supervised involved undergraduates. S.E.R. and P.H. wrote the original manuscript draft. All authors edited and improved the manuscript.

## Competing interests
The authors declare no competing interests.
