## [Peer Review File · Nature Communications]

Full title: Hydrogen and dark oxygen drive microbial productivity in diverse groundwater ecosystemsReviewer #1 (Remarks to the Author):

This manuscript compiles a substantial body of work that will be of interest to many. Noteworthy results include the discovery of higher cell densities in deeper, older groundwater, especially when complex organic material (in shales and coal beds for instance) are present. In addition, the discovery that there is greater O₂ present than previously considered possible is of note, and draws in to question previous assumptions that the deeper the subsurface, the more anoxic it is.

The research underpinning this manuscript is scientifically sound. The authors bring together geochemical, microbiological and genomic data in an elegant way. However, I have major concerns over the soundness of the conclusions drawn about metabolic activity from predominantly 16S rRNA sequencing data.

Although the authors acknowledge the limitations of making such metabolic inferences from taxonomic data, I do not think they have enough evidence to show "geochemistry and microbial ecology were closely linked". Whilst they present a well written argument for why they draw these inferences, and cite ample prior studies that can back up their assumptions, fundamentally you cannot robustly predict metabolic potential or even activity from 16S rRNA sequencing. How confident are you that the lineages you cite as capable of "aerobic and anaerobic hydrogen, methane, nitrogen and sulfur cycling" are what you have in your datasets? Genus-level taxonomic assignment alone is coarse, and many of these genera on which these inferences are made are metabolically-diverse. Just because prior research has demonstrated a member of a genus to fulfil function X does not mean all members of that genera can do this process. It is worthy of mention, but in this case much of the arguments in the manuscript are based on these types of assumptions, which I think is a stretch.

Interestingly, the authors carried out metagenomic sequencing on a subset of samples (though I could not find in the manuscript which samples these were and why they were chosen). Why not test some of the 16S rRNA-based assumptions using metagenomic data?

On the subject of the metagenomic data, the authors report genes implicated with chlorite dismutation being "moderately abundant" in the older / deeper groundwaters, but compared to what? Did you screen for these genes in other sample types too?

Aside from these specific concerns, I feel the authors could have done more to help the reader navigate the study (which currently would be difficult to reproduce). For example, it is not clear what the abbreviations used in the Supplementary Data files are. Why not add a second tab which defines these? Also, from the methods section, whilst I see all the tools they used for metagenomic bioinformatics approaches, it is not clear exactly how they calculate abundance. The supplementary data files were also not correctly labelled - the text points to Data S5 for metagenomic sequence analyses, but the file assigned to this name is an ASV table, presumably from 16S rRNA sequencing data.

On the whole, this is a major body of work with broad interest to the environmental microbiology community, detailing some interesting findings. However, from the submitted manuscript in its current state, I am not fully convinced that their microbial ecology conclusions are sound from the data they draw upon. With some modifications to the language, and / or expansion of the analysis they could do with their metagenomic data to validate their assumptions, this could be dealt with.

Reviewer #2 (Remarks to the Author):

In the manuscript "Hydrogen and dark oxygen drive microbial productivity in diverse groundwater ecosystems", Ruff et al. investigated the geochemistry and microbiology of

138 groundwater samples from 87 monitoring wells in the Canadian Prairie. A higher number of microbial cells were unexpectedly observed in older than in younger groundwaters. Both hydrogen and methane were presented in quite some of these samples. Isotope analysis suggested that there was potential dark oxygen formation via NO or chlorite dismutation processes. 16S rRNA amplicon sequencing and metagenomic analysis indicated the presence of key hydrogen and methane metabolizing microorganisms, as well as potential NO or chlorite dismutation bacteria. The findings are very interesting, expanding the current understanding of the microbiology of aquifer ecosystems. The story would be more robust if the authors could demonstrate hydrogen- and methane-metabolizing activity for some of the groundwater samples, though the current state of the manuscript is publishable in my opinion.

Minor comments:

- 1. The authors found a number of chlorite dismutation microbes presented in the studied aquifers. Were chlorite and chlorate detected, (if so, how much) in the groundwater samples?**
- 2. What's the similarity between the Hydrotalea nod sequence assembled in this study and other Methyloirabilis nod sequences? Does it also possess the characteristic Nod AA substitutions?**
- 3. The resolution of supplementary figures, such as Fig S8,9 is too low.**
- 4. Line 173, delete 'sequence'**

REVIEWER COMMENTS

Reviewer #1 (Remarks to the Author):

This manuscript compiles a substantial body of work that will be of interest to many. Noteworthy results include the discovery of higher cell densities in deeper, older groundwater, especially when complex organic material (in shales and coal beds for instance) are present. In addition, the discovery that there is greater O₂ present than previously considered possible is of note, and draws in to question previous assumptions that the deeper the subsurface, the more anoxic it is.

The research underpinning this manuscript is scientifically sound. The authors bring together geochemical, microbiological and genomic data in an elegant way. However, I have major concerns over the soundness of the conclusions drawn about metabolic activity from predominantly 16S rRNA sequencing data.

We thank the reviewer for the overall enthusiasm about our work and the helpful suggestions. We agree with the above assessment and have made substantial revisions to the text, as well as provide new analyses of amplicon and metagenomic data. Please find detailed replies below, answering the specific questions posed. The line numbers refer to the resubmitted manuscript version without track changes. A version with track changes is however provided to evaluate all changes made to the original submission.

Although the authors acknowledge the limitations of making such metabolic inferences from taxonomic data, I do not think they have enough evidence to show "geochemistry and microbial ecology were closely linked".

We have rephrased this statement. It now reads L27ff: "Geochemistry and microbial ecology showed consistent trends suggesting large-scale aerobic and anaerobic hydrogen, methane, nitrogen, and sulfur cycling carried out by diverse microbial communities."

We have also rephrased other similar statements throughout the text to tone down, e.g., L68f: "[...] of investigating the biogeochemistry and microbial ecology of a broad range of aquifer environments."

L72f: "[...] whether geochemistry is consistent with microbial community composition [...]"

L153ff: "[...] close relatives of obligate hydrogenotrophic methanogens including *Methanobacterium*, *Methanoregula* and *Methanospirillum* showed high relative abundances of 16S rRNA gene amplicon sequence variants [...]"

Whilst they present a well written argument for why they draw these inferences, and cite ample prior studies that can back up their assumptions, fundamentally you cannot robustly predict metabolic potential or even activity from 16S rRNA sequencing. How

confident are you that the lineages you cite as capable of "aerobic and anaerobic hydrogen, methane, nitrogen and sulfur cycling" are what you have in your datasets?

As the reviewer correctly pointed out we cannot be 100 % confident of the lineages' metabolic capabilities, particularly when using short read amplicons. We have thus carried out further sequence analyses:

1. We added 3 additional metagenomes to have a total of 5 metagenomes from the groundwater wells GW114, GW144, GW218, GW265, GW972. We chose these wells because of their microbiology and geochemistry.
2. We added genome-centric analyses in addition to the gene-centric analyses.
3. We assembled 61 metagenome-assembled genomes (MAGs) and analyzed their metabolic capabilities using a comprehensive set of 450 genes.
4. We compared the community composition based on metagenome-derived 16S rRNA genes to the composition based on amplicon sequence variants (ASV).

The following list describes the new **gene-centric** analyses in more detail:

We have analyzed the metagenomic data of 5 samples using the software tool phyloFlash:

- a. phyloFlash maps metagenome-derived short reads against the SILVA 16S rRNA reference database. Short reads that derive from 16S rRNA genes present in the metagenome align to the SILVA taxa and are referred to as nearest taxonomic units (NTUs). They are basically metagenome-derived sequence variants, like ASVs. We used high quality NTUs (long alignment length, high sequence similarity) to perform community analyses and show that the resulting community composition is very similar to the community composition of the original ASV data (genus-level) as well as to the community composition of ASV data at species-level (new re-analysis of the ASV dataset). See Supplementary Figure 12.
- b. phyloFlash also reconstructs (near) full-length 16S rRNA gene sequences (rec16S) from shotgun metagenomes, a list is provided in Supplementary Data 6. The quality of the rec16S from all 5 samples was very high, because most sequences were reconstructed at high % sequence identity (column F), long alignment lengths (column G), and optimal e-values (column H) by both methods (EMIRGE and Spades) that phyloFlash uses. The taxonomy of these rec16S sequences is very similar to that of the ASVs, and of the NTUs, and comprises all the key players that we find in ASV and NTU datasets including *Methylobacter*, *Methylothera*, *Hydrogenophaga*, *Thiobacillus*.
- c. To verify the similarity between ASVs and rec16S we performed a blast analysis in which we used all ASV sequences and searched them against all high quality rec16S. We show that many ASV of our original study, including all key lineages with very high relative sequence abundances, shared over 99% often 100% sequence identity with the reconstructed full-length archaeal and bacterial 16S rRNA gene sequences. The results of this blastn analysis are listed in the new Supplementary Data 7 and are visualized in the new Supplementary Figure 13.

- d. phyloFlash analyses were added to the updated methods section in L588ff: “Metagenomic short reads were mapped to the SILVA SSU reference database to assign nearest taxonomic units, as well as full-length 16S/18S rRNA gene sequences were reconstructed from metagenomes using phyloFlash v3.4 and compared to ASVs using blastn.”
- ➔ Based on the phyloFlash and blastn results we conclude that the amplicon-based community composition across the studied 138 samples is a good representation of the actual community. And that the trends and the presence of lineages and guilds holds true even when investigating species-level ASV, species-level NTU, and reconstructed full-length 16S rRNA gene sequences.
 - ➔ We have added these insights to the main text in L211ff: “Analyses of metagenomes from five wells (GW114, GW144, GW218, GW265, and GW972, all bearing old groundwaters) corroborated the community composition detected via metabarcoding. Mapped metagenomic 16S rRNA short reads (Supplementary Fig. 12), as well as reconstructed full-length 16S rRNA gene sequences (Supplementary Data 6) showed the same key genera and often the same species as the ASV datasets. The reliability of the ASV-based community composition was further supported by very high sequence similarities between ASVs and reconstructed full-length 16S rRNA genes (Supplementary Fig. 13, Supplementary Data 7). Both sequencing approaches show the same key players, including *Methylobacter tundripaludum* [...], *Methylothermobacter* sp. [...], *Hydrogenophaga* sp. [...], *Sulfuricurvum* sp. [...] and *Thiobacillus* sp. [...]”

We have searched the five metagenomes for genes encoding nitric oxide dismutases (*nod*) and chlorite dismutases (*clt*). We found 5 additional *clt* genes some of them at considerable sequencing depths of 207x and 212x.

- ➔ We have added this result in L328ff: “We found 13 chlorite dismutase genes mainly from *Dechloromonas*, *Nitrospina*, *Nakamurella* and *Magnetospirillum* species in old, anoxic/hypoxic (DO: 0-0.55 mg L⁻¹) groundwaters of GW144, GW218, GW265, and GW972 (Supplementary Data 9). The chlorite dismutase genes were detected at considerable sequencing depths ranging from 4 to 212x, indicating a moderate to high abundance of the organisms having this gene. Particularly two chlorite dismutases assigned to *Magnetospirillum* had very high sequencing depth (207x and 212x).”

Genus-level taxonomic assignment alone is coarse, and many of these genera on which these inferences are made are metabolically diverse. Just because prior research has demonstrated a member of a genus to fulfil function X does not mean all members of that genera can do this process. It is worthy of mention, but in this case much of the arguments in the manuscript are based on these types of assumptions, which I think is a stretch.

We agree that activity cannot be inferred from taxonomy, especially when investigating the community at genus level. Species-level taxonomic assignment of ASVs was not

shown in the original study, because of the limitations of phylogenetic resolution using short 400 base pair amplicons. However, having full-length 16S rRNA gene sequences available now (Supplementary Data 6), we compared the subset of ASVs that was annotated to species level with the reconstructed full-length 16S rRNA sequences that were annotated at species level (Supplementary Figure 12). In the latter case the species-level annotation is very reliable. We found that the congruency was very high. For example, one of the key players in all aquifer datasets was *Methylobacter tundripaludum*, several ASVs (annotated to species level by the tool DADA2) had 100% sequence identity at very high confidence with several reconstructed full-length 16S rRNA sequences that were classified as *M. tundripaludum* (Supplementary Data 7).

Interestingly, the authors carried out metagenomic sequencing on a subset of samples (though I could not find in the manuscript which samples these were and why they were chosen). Why not test some of the 16S rRNA-based assumptions using metagenomic data?

We have originally focused on the *nod* (nitric oxide dismutase) and *cld* genes (chlorite dismutase) within the metagenomic dataset of the samples GW114 and GW144. As suggested by reviewer 1 we now have expanded the metagenomic analyses, both in depth as well as in the number of analyzed samples. In addition to verifying community composition using metagenome-derived 16S data we have also tested the 16S-based assumptions using metagenome-derived metabolic gene annotations. We include 61 high-quality metagenome-assembled genomes (MAGs; >90% completeness, <3% contamination) from metagenomes of the samples GW114, GW144, GW218, GW265 and GW972 (Supplementary Data 5). Each of these MAGs was screened for 450 genes diagnostic for major housekeeping, anabolic and catabolic pathways. A summary of selected genes and pathways of 32 MAGs is shown in the new Figure 7.

The following paragraphs describes the new **genome-centric** analyses in more detail:

Among the MAGs were four MAGs of a *Methylobacter* species (very likely *Methylobacter tundripaludum*, which was highly abundant in the reconstructed 16S rRNA gene dataset, as well as the amplicon dataset) that had all the necessary genes for aerobic methane oxidation, nitrogen fixation and general housekeeping pathways including glycolysis. The metagenomes also contained MAGs of four other methanotrophs (including *Methylicorpusculum*). To thrive on methane these organisms must activate the methane with molecular oxygen (O₂), even if some of these MAGs showed the potential to use nitrite as an alternative electron acceptor for the downstream oxidation of methanol to carbon dioxide. The critical aerobic activation step of methane is catalyzed by methane monooxygenases for which we find the necessary genes (*pmoABC*, *smmo/mmoDYZ*) in all methanotroph MAGs. This is shown in the new Figure 7. In addition, we find methylotrophic *Methylotenera* (five MAGs), hydrogen/sulfur-oxidizing *Hydrogenophaga* (4 MAGs), and sulfur-oxidizing *Thiobacillus* and *Sulfuricurvum* (one MAG each). The latter MAGs contained genes for the complete oxidation of sulfide to sulfate (*sqr/sox/fcc* systems) using oxygen and/or nitrate as an

electron acceptor. We found additional sulfur-oxidizers (Fig. 7), as well as one *Nitrosomonas* MAG with a partial pathway for the obligate aerobic ammonium oxidation.

We have discussed the metagenome-derived metabolic capabilities at several sections in the manuscript:

L189ff: “Shotgun metagenomics supported the inferences derived from amplicon sequencing and revealed genes encoding Ni-Fe hydrogenases, hydrogenotrophic methanogenesis and sulfate reduction in the groundwater communities (Fig. 7). A high-quality metagenome-assembled genome (MAG) of a population within the *Methanobacteriaceae* (UBA349, MAG-32) contained a complete methyl-coenzyme M reductase (*mcrABCG*) and all other genes needed for hydrogenotrophic methanogenesis (*cdhA-E*, *ptr*, *mch*, *fae*), yet lacked genes diagnostic for methylotrophic, or acetoclastic methanogenesis (Supplementary Data 5). A complete sulfate reduction pathway was found in MAG-18 an organism in the genus-level clade UBA2262 within the *Desulfovibrionaceae* (Fig. 7). The hydrogen could be derived from microbial fermentation, e.g., via Fe-Fe hydrogenases that were present in three MAGs affiliating with *Bacteroidia* (*Lutibacter*, UBA6024, JAAYJT01; Supplementary Data 5). These MAGs also encoded the TonB/SusC transport system, which many *Bacteroidia* use to thrive on oligosaccharides.”

L218ff: “Both sequencing approaches show the same key players, including *Methylobacter tundripaludum* (MAG-01 – 04), *Methylotenera* sp. (MAG-09 – 13), *Hydrogenophaga* sp. (MAG-14 – 17), *Sulfuricurvum* sp. (MAG-19) and *Thiobacillus* sp. (MAG-20; Fig. 7). ”

L222ff: “We detected *Hydrogenophaga* at high relative abundances in most aquifers (Fig. 6b, 7, 8a). *Hydrogenophaga* MAGs contained genes encoding hydrogenases, oxidases, and RuBisCO, but also a complete sulfur oxidation pathway (*sqr*, *fccAB*, *soeABC*, *soxABX*), a complete denitrification pathway (in 2 of 4 MAGs) and an aerobic carbon monoxide dehydrogenase (*coxSML*), suggesting a facultatively anaerobic lithoautotrophic lifestyle.”

L230ff: “[*Methylobacter* and *Methylotenera*] were also abundant in the metagenomes of the five wells and were represented by nine MAGs. *Methylobacter* MAGs had complete pathways for the oxidation of methane to carbon dioxide including the critical activation of methane with molecular oxygen via the particulate methane monooxygenase (*pmoABC*). It was recently suggested that *Methylobacter* can couple methane oxidation to denitrification in hypoxic conditions. Indeed, we found these organisms to be capable of partial denitrification, from nitrite to nitrous oxide. *Methylotenera* MAGs contained genes to oxidize methanol to carbon dioxide and reduce nitrate to nitrite (Fig. 7). Denitrification by *Methylotenera* was reported previously and hence *Methylotenera* could provide nitrite to *Methylobacter*, while *Methylobacter* provides methanol to *Methylotenera*. Complementary interactions between *Methylobacter* and *Methylotenera* were observed in a methane-rich shallow aquifer, a lake as well as in methylotrophic microbiomes of mesocosms. The metabolic capabilities of the consortium however suggest that oxygen needs to be available to

catalyze the initial activation of methane, even if subsequent steps can be coupled to denitrification. *Methylicorpusculum* sp. (MAG-05), in contrast seemed to be entirely lacking the capability to use other electron acceptors than oxygen (Fig. 7).”

L248ff: “We also retrieved three high-quality MAGs of the uncultured genus JABFRC01 within the *Methylomonadaceae* (MAG-06 – 08). These MAGs encoded a soluble and a particulate methane monooxygenase and complete methane oxidation pathway revealing that the organism is an aerobic methane oxidizer requiring oxygen for the initial oxidation of methane to methanol. Like the closely related *Methylobacter* it may use nitrite as an alternative electron acceptor for the further oxidation of methanol to carbon dioxide in hypoxic conditions (Fig. 7, Supplementary Data 5).”

L263ff: “*Thiobacillus* (MAG-20) was able to oxidize sulfide and thiosulfate to sulfate using oxygen, but also had a complete nitrate reduction pathway (Fig. 7). A similar lifestyle was likely for PFJX01 within the *Thiobacillaceae* (MAG-21), *Sulfurimicrobium* (MAG-22), *Rhodoferax* (MAG-23, -24), UBA2250 (MAG-26, -27) and ETT8 (MAG-29) affiliating with *Rhodocyclaceae* and *Rhodobacteraceae*.”

L272ff: “We found obligate aerobic ammonium oxidizing *Nitrosomonas*, whose MAG encoded a partial *pmoA/amoA* and a *hao* to oxidize ammonium to nitric oxide (Supplementary Data 5). Like the activation of methane, the ammonium monooxygenase requires molecular oxygen. The biomass produced by the diverse, coexisting autotrophs in turn may support multitudes of archaeal and bacterial heterotrophs, including DPANN archaea and *Patescibacteria*. Numerous MAGs represented *Flavobacteriales*, *Cytophagales*, *Chitinophagales* and *Bacteroidales*, heterotrophs with an appetite for oligosaccharides and the necessary transport systems.”

The methods section was updated to include additional metagenome analyses in L615ff: “Reads from each sample were mapped to each of the 5 samples using BMap and sequencing depth profiles were generated using the ‘jgi_summarize_bam_contig_depths’ function from MetaBat2. Assembled contigs were binned into metagenome-assembled-genomes (MAGs) using MetaBat2, CONCOCT and MaxBin2. DAS-Tool was used to integrate MAGs produced by the three binning tools. The MAG properties including completeness and contamination were assessed by CheckM v1.2.0. To estimate the relative abundance of MAGs we determined the percentage of reads that mapped to each bin and the percentage that mapped to unbinned contigs using *checkm coverage* and *checkm profile* and then calculated ($\% \text{ reads mapped to bin} \times (100 - (\% \text{ reads mapped to unbinned contigs}))$). MAGs were classified using GTDB-tk (version 2.1.0, database release r207). Metagenomic short reads were mapped to the SILVA SSU reference database to assign nearest taxonomic units, as well as full-length 16S/18S rRNA gene sequences were reconstructed from metagenomes using phyloFlash v3.4 and compared to ASVs using blastn. Transfer

RNA, ribosomal RNA, CRISPR elements, and protein-coding genes including nitric oxide dismutase (*nod*) and chlorite dismutase (*cld*) coding genes were predicted and annotated using MetaErg v2.2.x. Amino acid sequences on contigs were additionally searched against representative *nod* and *cld* sequences from NCBI using blastp (e-value $1e^{-10}$). Dismutase amino acid sequences were aligned using Clustal Omega.”

On the subject of the metagenomic data, the authors report genes implicated with chlorite dismutation being "moderately abundant" in the older / deeper groundwaters, but compared to what? Did you screen for these genes in other sample types too?

We screened the additional metagenomes and found another five *cld* genes. This means that *cld* genes were present in 4 out of 5 metagenomes (Supplementary Data 9). We used sequencing depth as a proxy for abundance and concluded that based on the sequencing depth of the gene, the respective organism is likely moderately abundant in the community. We rephrased this statement in L331ff: “The chlorite dismutase genes were detected at considerable sequencing depths ranging from 4 to 212×, indicating a moderate to high abundance of the organisms having this gene. Particularly two chlorite dismutases assigned to *Magnetospirillum* had very high sequencing depth (207× and 212×).”

Aside from these specific concerns, I feel the authors could have done more to help the reader navigate the study (which currently would be difficult to reproduce). For example, it is not clear what the abbreviations used in the Supplementary Data files are. Why not add a second tab which defines these?

We agree and thoroughly modified all Supplementary Data files, parameters are now explained, and the methods/samples color-coded. Particularly Supplementary Data 1 is now much more comprehensive, because all abbreviations are explained in the top row of all listed parameters. We also rewrote the Supplementary text to match the sections in the main manuscript and make Supplementary figures easier to find. To enable reproducibility of the figures we provide a folder containing the locations or files of all the raw data needed to reproduce the figures.

Also, from the methods section, whilst I see all the tools they used for metagenomic bioinformatics approaches, it is not clear exactly how they calculate abundance.

Indeed, we did not measure “abundance” of genes but “sequencing depth”, which is a proxy for abundance, but not abundance per se. We have rephrased this and the respective sections now read:

L329ff: “The chlorite dismutase genes were detected at considerable sequencing depths ranging from 4 to 212×, indicating a moderate to high abundance of the

organisms having this gene. Particularly two chlorite dismutases assigned to *Magnetospirillum* had very high sequencing depth (207× and 212×).”

L338ff: “The *nod* affiliating with *Sediminibacterium* was sequenced at a depth of 700× corroborating the very high abundance of *Sediminibacterium* in the community (12 %).”

The % abundance of MAGs was estimated using CheckM, which also assesses the quality of MAGs. Whenever we refer to the abundance of a MAG (e.g., in Figure 7) it is based on estimates of the relative abundance of each bin by adjusting for the size of each genome and assuming a specific average genome size for all unbinned populations. We have modified the methods section accordingly L620ff:”To estimate the relative abundance of MAGs we determined the percentage of reads that mapped to each bin and the percentage that mapped to unbinned contigs using *checkm coverage* and *checkm profile* and then calculated $(\% \text{ reads mapped to bin}) * (100 - (\% \text{ reads mapped to unbinned contigs}))$.”

The supplementary data files were also not correctly labelled - the text points to Data S5 for metagenomic sequence analyses, but the file assigned to this name is an ASV table, presumably from 16S rRNA sequencing data.

We have corrected this oversight and checked all Supplementary data files, including the new Supplementary Data files derived from metagenome analyses. We also took the liberty to remove the cell counts and oxygen consumption from former Supplementary Data 1 to make it less complex. The cell counts and related details are now shown in Supplementary Data 2, and the oxygen consumption estimates are shown in Supplementary Data 8.

On the whole, this is a major body of work with broad interest to the environmental microbiology community, detailing some interesting findings. However, from the submitted manuscript in its current state, I am not fully convinced that their microbial ecology conclusions are sound from the data they draw upon. With some modifications to the language, and / or expansion of the analysis they could do with their metagenomic data to validate their assumptions, this could be dealt with.

Thank you for the overall enthusiasm about our work. We have now added extensive analyses of 4 additional metagenomes.

- We show that the 16S rRNA gene-based composition of the metagenomes (Supplementary Fig. 12, Supplementary Data 6) is very similar to the metabarcoding based community composition, both on species level (Supplementary Fig. 12c), as well as on genus level (Supplementary Fig. 12d).
- We show with a blast analysis that the sequence similarity between ASVs and full-length 16S rRNA sequences reconstructed from the metagenome is very high (Supplementary Fig. 13, Supplementary Data 7).
- We assembled and annotated 61 high-quality metagenome-assembled genomes (>90% completeness, <5% contamination) from the 5 metagenomes (Supplementary Data 5). And used a set of 450 marker genes to investigate the

presence of housekeeping and structural genes, as well as of major energy-conserving pathways (Supplementary Data 5).

- We show that the key players that we found in the metabarcoding survey, are also abundant in the metagenome, and possess the pathways which we previously only assumed to be present based on amplicon data; pathways for the oxidation of methane and sulfur using oxygen or nitrate.
- In addition to the above-mentioned new figures and supplementary data, we have modified figure 10 (former figure 9) and added lime-green circles, showing for which postulated metabolic pathways we have metagenomic evidence

This finding added considerable depth to the study:

- We found a full methane oxidation pathway in *Methylobacter* MAGs, and full methanol degradation in *Methylotenera*. In addition to oxygen *Methylobacter* are able to use nitrite, while *Methylotenera* can use nitrate. It thus seems that the organisms metabolically complement each other with methanol for nitrite. Importantly, the *Methylobacter* MAGs possess the genes encoding all subunits of the methane monooxygenase (*pmoABC*). This is the first and critical step for the activation of methane and requires molecular oxygen, regardless of a potential further degradation of C1-compounds using nitrite. Without the initial aerobic activation of methane and oxidation to methanol the *Methylobacter/Methylotenera* consortium would not be able to thrive and likely not be as abundant as detected. We also found a methanotroph of the genus *Methylicorpusculum*, which lacked genes for denitrification and thus seemed to be dependent on oxygen as sole electron acceptor.

Reviewer #2 (Remarks to the Author):

In the manuscript “Hydrogen and dark oxygen drive microbial productivity in diverse groundwater ecosystems”, Ruff et al. investigated the geochemistry and microbiology of 138 groundwater samples from 87 monitoring wells in the Canadian Prairie. A higher number of microbial cells were unexpectedly observed in older than in younger groundwaters. Both hydrogen and methane were presented in quite some of these samples. Isotope analysis suggested that there was potential dark oxygen formation via NO or chlorite dismutation processes. 16S rRNA amplicon sequencing and metagenomic analysis indicated the presence of key hydrogen and methane metabolizing microorganisms, as well as potential NO or chlorite dismutation bacteria. The findings are very interesting, expanding the current understanding of the microbiology of aquifer ecosystems. The story would be more robust if the authors could demonstrate hydrogen- and methane-metabolizing activity for some of the groundwater samples, though the current state of the manuscript is publishable in my opinion.

We thank the reviewer for the positive evaluation and agree that activity data (e.g., metatranscriptomics, -proteomics, or biogeochemical rate measurements) are desirable. In this first study we set out to gain a broad overview of the processes occurring in

these aquifers on a very large geographical and geological scale. The geochemistry, sequencing and microscopy provide multiple lines of evidence for active C-, S-, and N-cycling metabolisms and communities. We have now planned to understand these processes, rates, and activities in a much smaller set of samples at high-resolution.

Minor comments:

1. The authors found a number of chlorite dismutation microbes presented in the studied aquifers. Were chlorite and chlorate detected, (if so, how much) in the groundwater samples?

Chlorate and chlorite were unfortunately not measured, but nitrite was measured and present in many aquifers. For future studies we will aim to include chlorate and chlorite measurements.

2. What's the similarity between the *Hydrotalea nod* sequence assembled in this study and other *Methylomirabilis nod* sequences? Does it also possess the characteristic Nod AA substitutions?

Yes, the *nod* that we found has the diagnostic AA substitutions. We now have included an amino acid alignment highlighting the similarities between the putative *nod* in our samples and the *nod* found in *M. oxyfera*, as well as the differences to *nor* genes from *Pseudomonas denitrificans* and *Paracoccus denitrificans*. The full alignment is included in the "Raw data" folder. A representative excerpt of the alignment is shown in the new Supplementary Figure 15. On a side note: We have re-analyzed and curated the MAG and it now affiliates with *Sediminibacterium*, a close relative in the same family than *Hydrotalea*. Both were shown to have a *nod* by a recent study (Murali, R., Hemp, J. & Gennis, R. B. Evolution of quinol oxidation within the heme-copper oxidoreductase superfamily. *Biochim. Biophys. Acta - Bioenerg.* **1863**, 148907 (2022)).

We have included the following statement in L340ff: "Nitric oxide reductase is related to *nod* but it can be identified and distinguished by several diagnostic amino acid residues in the active center of the enzyme (Supplementary Fig. 15)."

3. The resolution of supplementary figures, such as Fig S8,9 is too low.

We agree and have modified all figures in the main text and supplement to maximize font sizes and readability. The figure features are now larger and the resolution much higher.

4. Line 173, delete 'sequence'

We included the word "sequence" to assure that the reader realizes we are referring to sequence abundance, and not cell abundance or other measures. We have deleted the word "sequence", but have rephrased the sentence to retain the abundance information. It now reads (L182ff): "*Candidatus Desulforudis audaxviator* were the most abundant

sulfate reducers across all ASV datasets, yet predominantly occurred in old groundwaters.”

Reviewer #1 (Remarks to the Author):

The authors have substantially expanded their manuscript and analyses in response to my comments, and have produced an excellent, publication-ready piece of work as a result. I am satisfied that all of my comments have been addressed, and support publication of the revised version.

Reviewer #2 (Remarks to the Author):

It is good to see that all the comments and suggestions were adequately addressed, and the manuscript has been greatly improved. Well done!